# Naturally occurring substitution in one amino acid in VHSV phosphoprotein enhances viral virulence in flounder

Jee Youn Hwang[1], Unn Hwa Lee[2], Min Jin Heo[3], Min Sun Kim[4], Ji Min Jeong[1], So Yeon Kim[5], Mun Gyeong Kwon[1], Bo Young Jee[1], Ki Hong Kim[5]*, Chan-Il Park[3]*, Jeong Woo Park[2]*

1 Aquatic Disease Control Division, National Institute Fisheries Science, Busan, Korea, 2 Department of Biological Sciences, University of Ulsan, Ulsan, Korea, 3 Department of Marine Biology & Aquaculture, Institute of Marine Industry, College of Marine Science, Gyeongsang National University, Gyeongnam, Korea, 4 Department of Integrative Bio-industrial Engineering, Sejong University, Seoul, Korea, 5 Department of Aquatic Life Medicine, Pukyong National University, Busan, Korea

☯ These authors contributed equally to this work.
* khkim@pknu.ac.kr (KHK); vinus96@hanmail.net (CIP); jwpark@ulsan.ac.kr (JWP)

**Data Availability Statement:** All nucleotide sequences and amino acid sequences files are available from the Genbank database (accession number(s) KY979963, KY979962, KY979961,

## Abstract

Viral hemorrhagic septicemia virus (VHSV) is a rhabdovirus that causes high mortality in cultured flounder. Naturally occurring VHSV strains vary greatly in virulence. Until now, little has been known about genetic alterations that affect the virulence of VHSV in flounder. We recently reported the full-genome sequences of 18 VHSV strains. In this study, we determined the virulence of these 18 VHSV strains in flounder and then the assessed relationships between differences in the amino acid sequences of the 18 VHSV strains and their virulence to flounder. We identified one amino acid substitution in the phosphoprotein (P) (Pro55-to-Leu substitution in the P protein; $P^{P55L}$) that is specific to highly virulent strains. This $P^{P55L}$ substitution was maintained stably after 30 cell passages. To investigate the effects of the $P^{P55L}$ substitution on VHSV virulence in flounder, we generated a recombinant VHSV carrying $P^{P55L}$ (rVHSV-P) from rVHSV carrying P55 in the P protein (rVHSV-wild). The rVHSV-P produced high level of viral RNA in cells and showed increased growth in cultured cells and virulence in flounder compared to the rVHSV-wild. In addition, rVHSV-P significantly inhibited the induction of the IFN1 gene in both cells and fish at 6 h post-infection. An RNA-seq analysis confirmed that rVHSV-P infection blocked the induction of several IFN-related genes in virus-infected cells at 6 h post-infection compared to rVHSV-wild. Ectopic expression of $P^{P55L}$ protein resulted in a decrease in IFN induction and an increase in viral RNA synthesis in rVHSV-wild-infected cells. Taken together, our results are the first to identify that the P55L substitution in the P protein enhances VHSV virulence in flounder. The data from this study add to the knowledge of VHSV virulence in flounder and could benefit VHSV surveillance efforts and the generation of a VHSV vaccine.

KY979960, KY979957, KY979959, KY979958, KY979952, KY979956, KY979955, KY979954, KY979953, KY979951, KY979950, KY979949, KY979948, KY979947, KY979946).

**Funding:** This work was supported by a grant from the National Institute of Fisheries Science in the Republic of Korea (www.nifs.go.kr) (grant number: R2021071) to JWP and the National Research Foundation of Korea (www.nrf.re.kr) (NRF-2014R1A6A1030318) to JWP. The funders had no role in study design, data collection and analysis, decision to publish, or preparation of the manuscript.

**Competing interests:** The authors have declared that no competing interests exist.

## Author summary

Viral hemorrhagic septicemia virus (VHSV) is a rhabdovirus that causes huge economic losses to the fish culture industry throughout the world. Virulence among naturally occurring VHSV strains varies widely. However, little is known about the viral factors that determine VHSV virulence. Here, we identify a naturally-occurring, single-amino-acid substitution in the VHSV P protein that enhances VHSV virulence in flounder. This amino acid substitution in the P protein was detected only in highly virulent VHSV strains, and it enhances viral RNA synthesis and inhibits the interferon response of host cells early after virus infection. Recombinant VHSV containing this amino acid substitution caused increased mortality in flounder compared with the wild type. This is the first study to identify a naturally occurring amino acid substitution in VHSV that determines its virulence in flounder. We expect that our result can be applied to other fish species, and this finding will provide new opportunities to generate an effective VHSV vaccine.

## Introduction

Viral hemorrhagic septicemia virus (VHSV) is the etiological agent of viral hemorrhagic septicemia (VHS), which causes huge economic losses to the fish culture industry. VHSV is an enveloped negative-strand RNA virus in the genus *Novirhabdovirus* of the family *Rhabdoviridae* [1]. The VHSV genome consists of approximately 11,200 nucleotides and contains six genes that encode the nucleocapsid- (N), phospho- (P), matrix- (M), glyco-(G), non-virion (NV)- and RNA polymerase (L) protein [2]. Phylogenetic analyses of the N- and G-encoding genes of VHSV have identified four main genotypes (I–IV), with several subgroups within genotypes I (minimum Ia-Ie) and IV (IVa–IVb) [3–5]. VHSV was first isolated in rainbow trout in Europe [6], and it has since been isolated in approximately 80 different fish species worldwide [7, 8]. Since its first report in cultured olive flounder in 1996 (*Paralichthys olivaceus*) [9], VHSV has caused serious economic problems for olive flounder farming in Japan [10, 11] and Korea [12, 13]. All VHSV strains thus far identified in cultured flounder in Asia belong to the genotype IVa clade. VHSV strains show a wide spectrum of virulence in cultured flounder that ranges from non-apparent infection to severe systemic disease causing high mortality. However, our understanding of the viral factors responsible for VHSV virulence in flounder is limited.

The virulence of VHSV is host-specific and varies generally among the viral genotypes and sub-genotypes. European genotype Ia and Ic isolates are virulent for rainbow trout, whereas VHSV marine isolates of genotype Ib, II and III are nearly all avirulent or low virulence in rainbow trout [14–16]. On the other hand, genotype IV isolates are virulent in several wild marine and freshwater fish species, but they are not virulent in rainbow trout [17–20]. Many researchers have tried to identify the molecular basis of VHSV virulence. It has been reported that the viral G-protein produces neutralizing antibodies in fish [21–23] and plays an important role in determining the virulence of VHSV [24, 25]. It has also been suggested that a single amino acid substitution in the L protein of VHSV can change the virulence of VHSV to the rainbow trout gill epithelium [26]. In addition, a comparison of the amino acid sequences of viral proteins between selected virulent and avirulent VHSV isolates identified difference in 14 amino acids, of which five occurred in the N protein, three each in the P and L proteins, two in the G protein, and one in the NV protein [27, 28]. However, other studies using chimeric recombinants of VHSV revealed that G, NV, and L are not determinant of host-specific virulence in rainbow trout [29] and did not identify which viral protein is responsible for host-

specific virulence in rainbow trout [30, 31]. Thus, factors determining VHSV virulence remain to be determined.

The P protein of a rhabdovirus is a multifunctional protein that is indispensable for both viral replication and evasion of host innate immunity. Specifically, this protein plays an essential role in viral RNA synthesis as a cofactor of the viral RNA-dependent RNA polymerase (L protein) by bridging the N protein, which bind directly to the viral genomic RNA, and the L protein in the ribonucleoprotein complex [32]. In addition, the P protein antagonizes type I interferon (IFN)-mediated antiviral responses by inhibiting signaling pathways for both IFN induction and response [33–37]. The P protein suppresses the activation of interferon regulatory factor 3 (IRF- 3), which is an important transcription factor for IFN induction [33, 36]. It also binds to the transcriptional factors signal transducers and activator of transcription 1 (STAT1) and STAT2, which play a key role in the IFN response by activating the expression of IFN-stimulated genes (ISGs), and inhibits their nuclear translocation and DNA binding [34, 37].

Fish type I IFNs also induce the expression of a wide variety of ISGs, such as the *Mx*, IFN-stimulated gene 15 (*Isg15*), and protein kinase R (*Pkr*) genes and promote an antiviral state in fish cells [38, 39]. Type I IFN responses play an important role in defense against VHSV infection. For example, VHSV infection induces IFN and ISGs in olive flounder [40]. Pre-treating cells with a type I IFN stimulator prevents VHSV infection in flounder [41, 42]. This suggests that type 1 IFN induced by VHSV infection plays a critical role in driving an antiviral innate immune response in fish and that VHSV virulence could correlate with its ability to inhibit that IFN-induced innate immune response. However, it has not yet been determined which VHSV protein or amino acid substitution is involved in modulating the innate immune response in VHSV-infected cells.

In this study, to clarify the VHSV protein or amino acid alterations associated with VHSV virulence in olive flounder, we determined the olive flounder-virulence of 18 VHSV strains isolated from cultured olive flounder in Korea and undertook an amino acid sequence analysis of the N, P, M, G, NV, and L proteins of those 18 VHSV strains. We identified one amino acid substitution, P55L in the VHSV P protein (P$^{P55L}$), associated with virulence. To determine whether P$^{P55L}$ influences VHSV virulence, we used reverse genetics to generate recombinant VHSV containing this amino acid substitution while keeping the same genetic background in other regions. The P$^{P55L}$ substitution enhanced the ability of VHSV to block the host IFN response and synthesize viral RNA, and thus it enhanced the virulence of VHSV *in vitro* and *in vivo*. This is the first report that an amino acid substitution in the VHSV P protein enhances VHSV virulence in flounder, and this finding could provide new information about how amino acid alteration enhance VHSV virulence.

## Results

### Virulence of VHSV strains in olive flounder

Eighteen VHSV strains were isolated from cultured olive flounder during multiple outbreaks of VHS in Korea between 2012 to 2016 (S1 Table). All the strains are members of genotype IVa, and their full genome sequences have been reported [43]. To assess their virulence, olive flounder (body weight 31.85 ± 3.89 g) were intraperitoneally (i.p.) injected with a $1 \times 10^4$ plaque-forming units (PFU) of each viral strain. Following the i.p. injection with VHSV, the fish appeared healthy until day 4. The VHSV-infected fish showed typical VHS lesions on day 4 and began to die on day 5. Even though all 18 VHSV strains were isolated during the VHS outbreak, they varied in their virulence in olive flounder (Fig 1 and Table 1). Based on the average mortality observed in four independent experiments, we selected 7 VHSV strains that caused

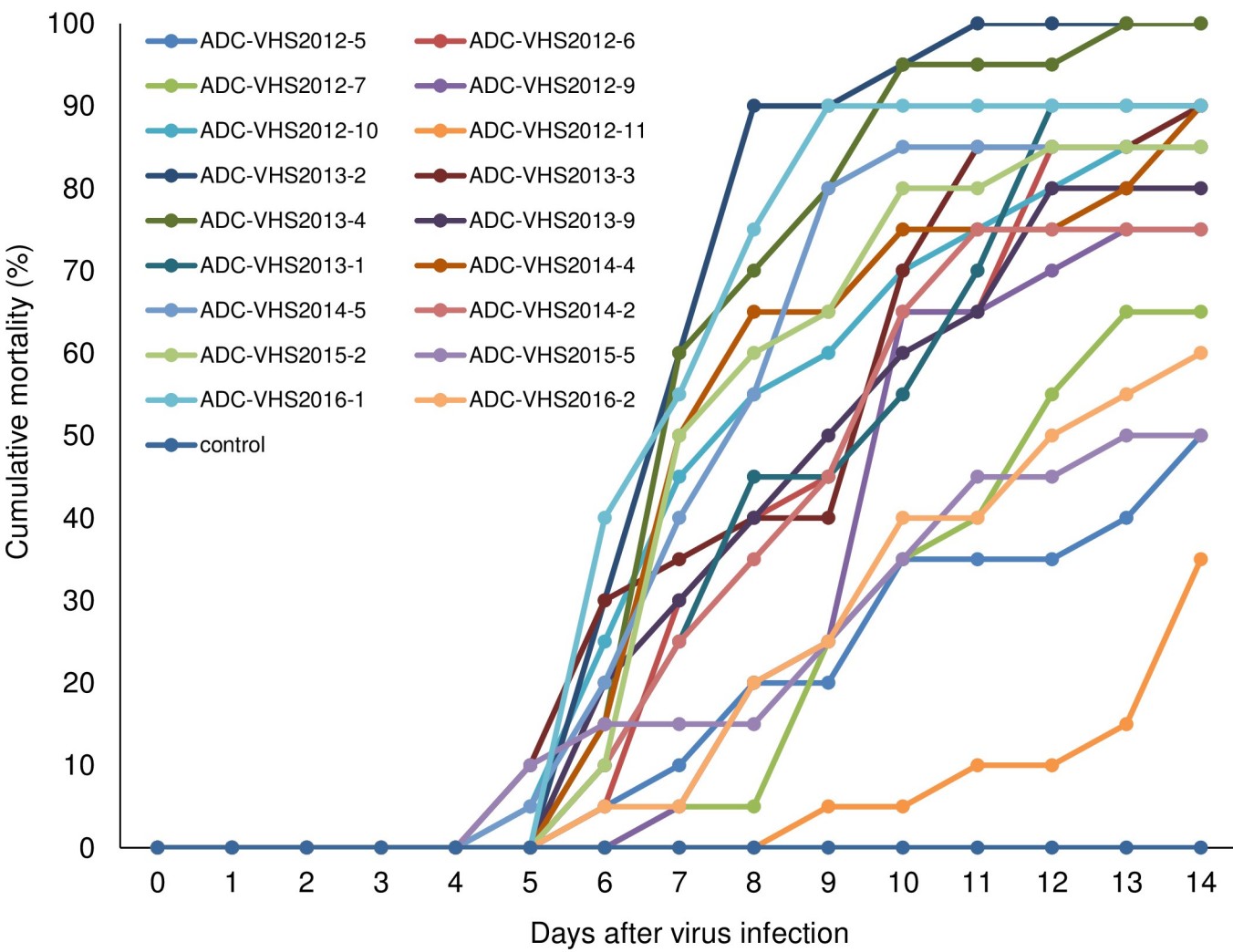

**Fig 1. Cumulative mortality of olive flounder infected with 18 VHSV strains.** Olive flounder weighing 31.85 ± 3.89 g (mean ± SD) were intraperitoneally injected with $1\times10^4$ $TCID_{50}$/fish/0.1 ml of VHSV at 11–13˚C. Fish injected with medium were used as the control. Fish mortality was monitored daily, and the graph was generated using the cumulative mortality from "Exp2" of four independent experiments in Table 1 (n = 20 per group).

70–75% cumulative mortality (high-virulence strains) and 6 VHSV strains that caused 31.25–57.5% cumulative mortality (low-virulence strains) in olive flounder (Fig 1 and Table 1). In the mock-infected negative control, mortality was recorded as 4%.

## Identification of amino acid sequence variations associated with high VHSV virulence in flounder

To identify amino acid changes associated with high VHSV virulence in olive flounder, we compared the amino acid sequences of the N, P, M, G, NV, and L proteins from the 18 VHSV strains and found variations in 95 amino acid residues: 14 residues in N, 6 residues in P, 10 residues in M, 14 residues in G, 7 residues in NV, and 44 residues in L (Tables 2–7). The number of amino acid sequence variations was thus highest in L, followed by G/N, M, NV, and P (Table 8). However, the ratio of amino acid sequence variation was highest in NV, followed by M, N, G, P, and L (Table 8), suggesting that the amino acid sequences of P and L are more conserved than those of the other proteins among the 18 VHSV strains. Among the 95 amino acid

**Table 1. Virulence of the 18 VHSV strains used in this study in olive flounder.**

| Virus strains | Mortality (%) | | | | | Virulence phenotype |
| | Exp1[*] | Exp2[**] | Exp3[**] | Exp4[**] | Average | |
|---|---|---|---|---|---|---|
| ADC-VHS2015-5 | 40 | 50 | 15 | 20 | 31.25 | low |
| ADC-VHS2012-11 | 40 | 35 | 75 | 50 | 50.00 | low |
| ADC-VHS2012-7 | 90 | 65 | 40 | 15 | 52.50 | low |
| ADC-VHS2014-2 | 70 | 75 | 40 | 30 | 53.75 | low |
| ADC-VHS2016-2 | 70 | 60 | 35 | 60 | 56.25 | low |
| ADC-VHS2012-5 | 70 | 50 | 65 | 45 | 57.50 | low |
| ADC-VHS2012-10 | 60 | 85 | 50 | 45 | 60.00 | moderate |
| ADC-VHS2012-9 | 70 | 75 | 60 | 45 | 62.50 | moderate |
| ADC-VHS2015-2 | 40 | 85 | 60 | 65 | 62.50 | moderate |
| ADC-VHS2013-1 | 70 | 90 | 40 | 60 | 65.00 | moderate |
| ADC-VHS2013-3 | 80 | 90 | 55 | 45 | 67.50 | moderate |
| ADC-VHS2013-9 | 70 | 80 | 60 | 70 | 70.00 | high |
| ADC-VHS2013-2 | 70 | 100 | 40 | 70 | 70.00 | high |
| ADC-VHS2014-5 | 90 | 85 | 60 | 45 | 70.00 | high |
| ADC-VHS2016-1 | 70 | 90 | 55 | 70 | 71.25 | high |
| ADC-VHS2013-4 | 90 | 100 | 80 | 20 | 72.50 | high |
| ADC-VHS2012-6 | 90 | 85 | 55 | 65 | 73.75 | high |
| ADC-VHS2014-4 | 90 | 90 | 50 | 70 | 75.00 | high |

[*], 10 fish per VHSV strain

[**], 20 fish per VHSV strain

**Table 2. Nucleoprotein (N) amino acid variations in the 18 VHSV strains.**

| VHSV strain | Mortality in flounder (%) | N amino acid at position | | | | | | | | | | | | | |
| | | 30 | 42 | 43 | 47 | 67 | 71 | 79 | 102 | 127 | 270 | 345 | 368 | 374 | 386 |
|---|---|---|---|---|---|---|---|---|---|---|---|---|---|---|---|
| ADC-VHS2015-5 | 31.25 | G | G | A | K | V | F | E | T | I | R | K | A | P | G |
| ADC-VHS2012-11 | 50 | G | G | A | K | V | F | E | T | I | R | K | A | P | G |
| ADC-VHS2012-7 | 52.5 | G | G | A | K | V | F | G | T | I | R | K | A | P | G |
| ADC-VHS2014-2 | 53.75 | G | G | A | K | V | F | G | T | I | Q | K | A | P | G |
| ADC-VHS2016-2 | 56.25 | G | G | A | K | V | F | E | T | I | R | K | A | P | G |
| ADC-VHS2012-5 | 57.5 | G | D | T | N | I | Y | G | T | V | R | R | A | T | R |
| ADC-VHS2012-10 | 60 | G | G | A | K | V | F | G | T | I | R | K | A | P | G |
| ADC-VHS2012-9 | 62.5 | G | G | A | K | V | F | G | T | I | R | K | A | P | G |
| ADC-VHS2015-2 | 62.5 | G | G | A | K | V | F | G | T | I | R | K | A | P | G |
| ADC-VHS2013-1 | 65 | G | D | T | N | I | Y | G | T | V | R | R | A | T | R |
| ADC-VHS2013-3 | 67.5 | G | G | A | K | V | F | G | T | I | R | K | A | P | G |
| ADC-VHS2013-9 | 70 | G | G | A | K | V | F | G | A | I | R | K | A | P | G |
| ADC-VHS2013-2 | 70 | G | G | A | K | V | F | G | T | I | R | K | A | P | G |
| ADC-VHS2014-5 | 70 | G | G | A | K | V | F | G | A | I | R | K | A | P | G |
| ADC-VHS2016-1 | 71.25 | D | G | A | K | V | F | G | A | I | R | K | T | P | G |
| ADC-VHS2013-4 | 72.5 | G | G | A | K | V | F | G | T | I | R | K | A | P | G |
| ADC-VHS2012-6 | 73.75 | G | G | A | K | V | F | G | A | I | R | K | A | P | G |
| ADC-VHS2014-4 | 75 | G | G | A | K | V | F | G | A | I | R | K | A | P | G |

**Table 3. Phosphoprotein (P) amino acid variations in 18 the VHSV strains.**

| VHSV strain | Mortality in flounder (%) | P amino acid at position | | | | | |
| --- | --- | --- | --- | --- | --- | --- | --- |
| | | 2 | 22 | 42 | 55 | 69 | 209 |
| ADC-VHS2015-5 | 31.25 | T | D | S | P | D | M |
| ADC-VHS2012-11 | 50 | T | D | S | P | D | M |
| ADC-VHS2012-7 | 52.5 | T | D | S | P | D | M |
| ADC-VHS2014-2 | 53.75 | T | D | S | P | D | M |
| ADC-VHS2016-2 | 56.25 | T | D | P | P | D | M |
| ADC-VHS2012-5 | 57.5 | T | D | S | P | D | T |
| ADC-VHS2012-10 | 60 | T | D | S | L | D | M |
| ADC-VHS2012-9 | 62.5 | T | D | S | L | D | M |
| ADC-VHS2015-2 | 62.5 | T | D | S | L | E | M |
| ADC-VHS2013-1 | 65 | T | D | S | P | D | T |
| ADC-VHS2013-3 | 67.5 | T | D | S | L | D | M |
| ADC-VHS2013-9 | 70 | T | K | S | L | D | M |
| ADC-VHS2013-2 | 70 | T | D | S | L | D | M |
| ADC-VHS2014-5 | 70 | A | D | S | L | D | M |
| ADC-VHS2016-1 | 71.25 | A | D | S | L | D | M |
| ADC-VHS2013-4 | 72.5 | T | D | S | L | D | M |
| ADC-VHS2012-6 | 73.75 | T | D | S | L | D | M |
| ADC-VHS2014-4 | 75 | T | K | S | L | D | M |

sequence variations, we focused on those common in all 7 high-virulence VHSV strains but not in the 6 low-virulence VHSV strains. Using that criterion, we identified one amino acid substitution from proline at position 55 of the P protein in the low-virulence VHSV strains to leucine at the same amino acid residue of P the protein in the high-virulence VHSV strains.

**Table 4. Matrix protein (M) amino acid variations in the 18 VHSV strains.**

| VHSV strain | Mortality in flounder (%) | M amino acid at position | | | | | | | | | |
| --- | --- | --- | --- | --- | --- | --- | --- | --- | --- | --- | --- |
| | | 3 | 7 | 12 | 39 | 77 | 132 | 186 | 189 | 192 | 198 |
| ADC-VHS2015-5 | 31.25 | L | K | V | S | A | S | Q | P | R | V |
| ADC-VHS2012-11 | 50 | P | K | V | P | A | S | K | P | R | V |
| ADC-VHS2012-7 | 52.5 | L | K | V | P | A | S | Q | P | R | A |
| ADC-VHS2014-2 | 53.75 | L | K | V | P | A | S | Q | P | R | V |
| ADC-VHS2016-2 | 56.25 | L | K | V | P | A | S | Q | P | R | V |
| ADC-VHS2012-5 | 57.5 | L | K | V | P | A | N | H | L | K | V |
| ADC-VHS2012-10 | 60 | L | R | V | P | A | S | Q | P | R | V |
| ADC-VHS2012-9 | 62.5 | L | K | V | P | A | S | Q | P | R | V |
| ADC-VHS2015-2 | 62.5 | L | R | I | P | A | S | Q | P | R | V |
| ADC-VHS2013-1 | 65 | L | K | V | P | A | N | H | L | K | V |
| ADC-VHS2013-3 | 67.5 | L | K | V | P | A | S | Q | P | R | V |
| ADC-VHS2013-9 | 70 | L | K | V | P | A | S | Q | P | R | V |
| ADC-VHS2013-2 | 70 | L | K | V | P | A | S | Q | P | R | V |
| ADC-VHS2014-5 | 70 | L | K | V | P | S | S | Q | P | R | V |
| ADC-VHS2016-1 | 71.25 | L | K | V | P | S | S | Q | P | R | V |
| ADC-VHS2013-4 | 72.5 | L | R | V | P | A | S | Q | P | R | V |
| ADC-VHS2012-6 | 73.75 | L | K | V | P | A | S | Q | P | R | V |
| ADC-VHS2014-4 | 75 | L | K | V | P | A | S | Q | P | R | V |

**Table 5. Glycoprotein (G) amino acid variations in 18 the VHSV strains.**

| VHSV strain | Mortality in flounder (%) | G amino acid at position | | | | | | | | | | | | | |
|---|---|---|---|---|---|---|---|---|---|---|---|---|---|---|---|
| | | 10 | 12 | 52 | 71 | 136 | 184 | 249 | 272 | 284 | 339 | 380 | 383 | 385 | 493 |
| ADC-VHS2015-5 | 31.25 | T | V | N | T | N | N | M | T | K | R | I | S | N | S |
| ADC-VHS2012-11 | 50 | I | V | N | T | N | N | M | T | K | S | I | S | N | S |
| ADC-VHS2012-7 | 52.5 | I | V | N | T | N | N | M | T | K | S | I | S | N | S |
| ADC-VHS2014-2 | 53.75 | I | V | N | I | N | N | M | I | K | S | I | S | N | S |
| ADC-VHS2016-2 | 56.25 | I | V | N | T | N | N | M | T | K | S | I | S | K | S |
| ADC-VHS2012-5 | 57.5 | I | I | N | T | N | D | M | T | R | S | T | P | N | S |
| ADC-VHS2012-10 | 60 | I | V | N | I | N | N | M | T | K | S | T | S | N | S |
| ADC-VHS2012-9 | 62.5 | I | V | N | I | S | N | M | T | K | S | I | S | N | S |
| ADC-VHS2015-2 | 62.5 | I | V | N | I | N | N | M | T | K | S | T | S | N | S |
| ADC-VHS2013-1 | 65 | I | I | N | T | N | D | K | T | R | S | T | P | N | S |
| ADC-VHS2013-3 | 67.5 | I | I | N | I | N | N | M | T | K | S | I | S | N | G |
| ADC-VHS2013-9 | 70 | I | V | N | I | N | N | M | T | K | S | I | S | N | S |
| ADC-VHS2013-2 | 70 | V | V | N | I | N | N | M | T | K | S | I | S | N | S |
| ADC-VHS2014-5 | 70 | I | V | N | I | N | N | M | T | K | S | I | S | N | S |
| ADC-VHS2016-1 | 71.25 | I | V | D | I | N | N | M | I | K | S | I | S | N | S |
| ADC-VHS2013-4 | 72.5 | I | V | N | I | N | N | M | T | K | S | T | S | N | S |
| ADC-VHS2012-6 | 73.75 | I | I | N | I | N | N | M | T | K | S | I | S | N | S |
| ADC-VHS2014-4 | 75 | I | V | N | I | N | N | M | T | K | S | I | S | N | S |

This amino acid substitution, $P^{P55L}$, was detected in all 7 high-virulence VHSV strains but not in any of the 5 low-virulence VHSV strains (Table 3). We also detected two more amino acid substitutions that met our criterion: a threonine/isoleucine substitution at position 71 of the G protein, $G^{T71I}$, and a glutamine/arginine substitution at position 1079 of the L protein, $L^{Q1079R}$. However, even though all 7 high-virulence VHSV strains had those two amino acid

**Table 6. Non-virion (NV) protein amino acid variations in 18 the VHSV strains.**

| VHSV strain | Mortality in flounder (%) | NV amino acid at position | | | | | | |
|---|---|---|---|---|---|---|---|---|
| | | 8 | 56 | 81 | 88 | 117 | 119 | 120 |
| ADC-VHS2015-5 | 31.25 | S | S | T | V | G | E | S |
| ADC-VHS2012-11 | 50 | S | S | T | V | G | E | S |
| ADC-VHS2012-7 | 52.5 | S | S | T | V | G | E | S |
| ADC-VHS2014-2 | 53.75 | S | S | T | V | G | E | S |
| ADC-VHS2016-2 | 56.25 | S | S | T | V | G | E | S |
| ADC-VHS2012-5 | 57.5 | S | S | A | V | G | K | S |
| ADC-VHS2012-10 | 60 | S | S | T | V | G | E | S |
| ADC-VHS2012-9 | 62.5 | S | S | T | V | G | E | S |
| ADC-VHS2015-2 | 62.5 | S | S | T | V | G | E | P |
| ADC-VHS2013-1 | 65 | S | S | A | V | G | K | S |
| ADC-VHS2013-3 | 67.5 | S | L | T | V | D | E | S |
| ADC-VHS2013-9 | 70 | N | S | T | V | G | E | S |
| ADC-VHS2013-2 | 70 | S | S | T | V | G | E | S |
| ADC-VHS2014-5 | 70 | S | S | T | I | G | E | S |
| ADC-VHS2016-1 | 71.25 | S | S | T | V | G | E | S |
| ADC-VHS2013-4 | 72.5 | S | S | T | V | G | E | S |
| ADC-VHS2012-6 | 73.75 | S | S | T | V | G | E | S |
| ADC-VHS2014-4 | 75 | N | S | T | V | G | E | S |

**Table 7. L protein (L) amino acid variations in the 18 VHSV strains.**

| VHSV strain | Mortality in flounder (%) | 46 | 65 | 91 | 101 | 115 | 144 | 145 | 147 | 310 | 368 | 380 | 449 | 476 | 478 | 484 | 592 | 621 | 710 | 765 | 798 | 909 | 916 | 917 |
|---|---|---|---|---|---|---|---|---|---|---|---|---|---|---|---|---|---|---|---|---|---|---|---|---|
| | | | | | | | | | | | | | | | | | | **L amino acid at position** | | | | | | |
| ADC-VHS2015-5 | 31.25 | Y | K | G | V | K | D | V | G | A | T | G | S | K | D | K | G | V | H | R | V | V | K | K |
| ADC-VHS2012-11 | 50 | Y | K | G | V | K | D | V | G | A | T | G | S | K | D | K | G | M | H | R | V | V | K | K |
| ADC-VHS2012-7 | 52.5 | Y | K | G | V | R | D | V | G | A | T | S | S | K | D | K | G | V | H | R | A | V | K | K |
| ADC-VHS2014-2 | 53.75 | Y | K | G | V | K | D | V | G | A | T | G | S | K | D | K | G | V | H | R | V | I | K | K |
| ADC-VHS2016-2 | 56.25 | H | K | G | V | K | D | V | G | A | T | G | S | K | D | K | G | V | H | R | V | V | K | T |
| ADC-VHS2012-5 | 57.5 | Y | K | G | I | K | N | V | G | A | T | G | S | N | D | K | G | V | H | K | V | V | K | K |
| ADC-VHS2012-10 | 60 | Y | K | G | I | K | D | A | E | A | T | G | S | K | D | K | G | V | H | R | V | V | K | K |
| ADC-VHS2012-9 | 62.5 | Y | K | G | V | K | D | V | G | A | T | G | S | K | D | K | G | V | R | R | V | V | K | K |
| ADC-VHS2015-2 | 62.5 | Y | K | G | I | K | D | A | E | A | T | G | S | K | D | K | G | V | H | R | V | V | K | K |
| ADC-VHS2013-1 | 65 | Y | K | G | I | K | N | V | G | V | T | G | S | N | D | K | G | V | H | K | V | V | K | K |
| ADC-VHS2013-3 | 67.5 | Y | R | G | V | K | D | V | G | A | T | G | S | K | N | E | G | V | R | R | V | V | R | K |
| ADC-VHS2013-9 | 70 | Y | K | G | V | K | D | V | G | A | T | G | S | K | D | K | E | V | H | R | V | V | K | K |
| ADC-VHS2013-2 | 70 | Y | K | E | V | K | D | V | G | A | T | G | S | K | D | K | G | V | H | R | V | V | K | K |
| ADC-VHS2014-5 | 70 | Y | K | G | V | K | D | V | E | A | I | G | T | K | D | K | G | V | H | R | V | V | K | K |
| ADC-VHS2016-1 | 71.25 | Y | K | G | V | K | D | V | G | A | I | G | S | K | D | K | G | V | H | R | V | V | K | K |
| ADC-VHS2013-4 | 72.5 | Y | K | G | I | K | D | A | G | A | T | G | S | K | D | K | G | V | H | R | V | V | K | K |
| ADC-VHS2012-6 | 73.75 | Y | K | G | V | K | D | V | G | A | I | G | S | K | D | K | G | V | H | R | V | V | K | K |
| ADC-VHS2014-4 | 75 | Y | K | G | V | K | D | V | G | A | I | G | S | K | D | K | G | V | H | R | V | V | K | K |

| VHSV strain | Mortality in flounder (%) | 1056 | 1071 | 1075 | 1079 | 1087 | 1164 | 1176 | 1314 | 1368 | 1482 | 1485 | 1533 | 1536 | 1541 | 1552 | 1774 | 1802 | 1837 | 1841 | 1861 | 1936 |
|---|---|---|---|---|---|---|---|---|---|---|---|---|---|---|---|---|---|---|---|---|---|---|
| | | | | | | | | | | **L amino acid at position** | | | | | | | | | | | | |
| ADC-VHS2015-5 | 31.25 | Q | G | S | Q | Q | M | L | G | I | I | I | P | K | V | I | Q | V | I | Y | A | T |
| ADC-VHS2012-11 | 50 | Q | G | S | Q | Q | M | L | G | I | I | I | L | Q | V | I | Q | V | I | Y | A | T |
| ADC-VHS2012-7 | 52.5 | R | G | S | Q | Q | M | L | G | I | T | I | L | Q | V | I | Q | V | I | H | A | A |
| ADC-VHS2014-2 | 53.75 | Q | G | S | R | Q | M | L | G | I | I | I | L | Q | V | I | Q | V | I | Y | A | P |
| ADC-VHS2016-2 | 56.25 | Q | E | S | Q | Q | M | L | G | I | I | I | L | Q | V | I | Q | V | I | Y | A | T |
| ADC-VHS2012-5 | 57.5 | Q | E | S | Q | R | M | L | E | T | I | V | L | Q | V | I | Q | V | I | Y | A | T |
| ADC-VHS2012-10 | 60 | Q | G | S | R | Q | M | L | G | I | I | I | L | Q | V | I | R | V | I | Y | A | T |
| ADC-VHS2012-9 | 62.5 | Q | G | S | R | Q | M | L | G | I | I | I | L | Q | V | V | Q | V | I | Y | T | T |
| ADC-VHS2015-2 | 62.5 | Q | G | S | R | Q | M | P | G | I | I | I | L | Q | V | I | R | V | I | Y | A | T |
| ADC-VHS2013-1 | 65 | Q | E | S | Q | R | M | L | E | T | I | V | L | Q | V | I | Q | V | I | Y | A | T |
| ADC-VHS2013-3 | 67.5 | Q | G | L | R | Q | M | L | G | I | I | I | L | Q | V | I | Q | A | V | Y | A | T |
| ADC-VHS2013-9 | 70 | Q | G | S | R | Q | M | L | G | I | I | I | L | Q | V | I | Q | V | I | Y | A | T |
| ADC-VHS2013-2 | 70 | Q | G | S | R | Q | M | L | G | I | I | I | L | Q | V | I | Q | V | I | Y | A | T |
| ADC-VHS2014-5 | 70 | Q | G | S | R | Q | M | L | G | I | I | I | L | Q | V | I | Q | V | I | Y | A | T |
| ADC-VHS2016-1 | 71.25 | Q | G | S | R | Q | M | L | G | I | I | I | L | Q | V | I | Q | V | I | Y | A | T |
| ADC-VHS2013-4 | 72.5 | Q | G | S | R | Q | M | P | G | I | I | I | L | Q | V | I | R | V | I | Y | A | T |
| ADC-VHS2012-6 | 73.75 | Q | G | S | R | Q | M | L | G | I | I | I | L | Q | V | I | Q | V | I | Y | A | T |
| ADC-VHS2014-4 | 75 | Q | G | S | R | Q | L | L | G | I | I | I | L | Q | I | I | Q | V | I | Y | A | T |

substitutions, one low-virulence VHSV strain also had them (Tables 5 and 7). Our results thus suggest that three amino acid substitutions, $P^{P55L}$, $G^{T71I}$, and $L^{Q1079R}$, are associated with the virulence of VHSV in flounder, and among them, $P^{P55L}$ is most specifically associated with virulence.

## Generation of recombinant VHSVs with the of $P^{P55L}$, $G^{I71T}$, and $L^{Q1079R}$ amino acid substitutions

Previously, we generated a full-length cDNA clone (pVHSV-wild) (S1 Fig) and a recombinant VHSV (rVHSV-wild) from the VHSV KJ2008 strain, which contains the low-virulence amino

**Table 8. Amino acid variation number and ratio in the six proteins of the 18 VHSV strains.**

| | VHSV proteins | | | | | | |
|---|---|---|---|---|---|---|---|
| | **N** | **P** | **M** | **G** | **NV** | **L** | **Total** |
| Total amino acid residues | 404 | 222 | 201 | 507 | 122 | 1,984 | 3,440 |
| Number of amino acid variations | 14 | 6 | 10 | 14 | 7 | 44 | 95 |
| Amino acid variation ratio (%) | 3.47 | 2.70 | 4.98 | 2.76 | 5.74 | 2.22 | 2.76 |

acid residues, P, T, and Q, at position 55 of the P protein, 71 of the G protein, and 1079 of the L protein, respectively (GenBank accession no, JF792424) [44]. To determine the function of the three amino acid substitutions, we first performed site-directed mutagenesis to replace the proline (P) residue at position 55 of P with a leucine (L). By using plasmid-driven reverse genetics, we generated a recombinant VHSV, rVHSV-P$^{P55L}$ (hereafter rVHSV-P). Because the other two amino acid substitutions, G$^{T71I}$ and L$^{Q1079R}$, might also affect VHSV virulence, we created plasmids and recombinant VHSVs with the G$^{T71I}$ or G$^{T71I}$ and L$^{Q1079R}$ amino acid substitutions, in addition to P$^{P55L}$. We named these two additional rVHSVs rVHSV-PG (containing two amino acid substitutions, P$^{P55L}$ and G$^{T71I}$) and rVHSV-PGL (containing three amino acid substitutions, P$^{P55L}$, G$^{T71I}$, and L$^{Q1079R}$) (Fig 2A). The amino acid substitutions in each rVHSV were confirmed by RT-PCR and sequencing (Fig 2B).

## Effect of the P$^{P55L}$, G$^{T71I}$, and L$^{Q1079R}$ amino acid substitutions on virus growth in flounder cells

Highly virulent fish rhabdovirus grows more rapidly than low-virulence virus [45–47]. To determine whether these amino acid substitutions affected virus growth in flounder cells, we inoculated HINAE cells with recombinant VHSVs at 0.01 multiplicity of infection (MOI) at 14˚C. At 0, 1, and 3 days post-infection, titers of infectious virus released into the media were determined using a plaque assay. The rVHSV-P grew more efficiently than the rVHSV-wild, with the most notable difference in replication observed at 3 days post-infection: at day 3 post-infection, the viral titer of rVHSV-P was 9.7-fold higher than that of rVHSV-wild (Fig 2C). Thus the P$^{P55L}$ amino acid substitution enhanced the growth of VHSV in HINAE cells. We next determined whether the additional amino acid substitutions, G$^{T71I}$ and L$^{Q1079R}$, affected the virus growth in HINAE cells. As shown in Fig 2C, the viral titers of rVHSV-PG and rVHSV-PGL were similar to that of rVHSV-P, 9.2-fold and 9.9-fold higher than that of rVHSV-wild, respectively (Fig 2C). Overall, these results suggest that the P$^{P55L}$ amino acid substitution increases the growth of VHSV in HINAE cells and that the G$^{T71I}$ and L$^{Q1079R}$ amino acid substitutions have no additional effect on viral growth in HINAE cells.

## Stability of P$^{P55L}$ in VHSV during viral replication

To examine the stability of P$^{P55L}$ in VHSV, we serially passaged (30 times) a low-virulence VHSV strain (ADC-VHS2015-5) containing P$^{P55}$ and a high-virulence VHSV strain (ADC-VHS2012-6) containing P$^{P55L}$ in HINAE cells. Sequence analysis of the P gene encompassing amino acid residue 55 after passages P15, P20, P25, and P30 revealed that both the low-virulence ADC-VHS2015-5 strain and the high-virulence ADC-VHS2012-6 strain stably maintained P$^{P55}$ and P$^{P55L}$, respectively, after 30 passages (S1 Fig).

## Effects of the P$^{P55L}$ amino acid substitution on viral virulence in olive flounder

Because the P$^{P55L}$ amino acid substitution enhanced viral growth in cells (Fig 2C), we tested whether it would also affect virulence in flounder. To evaluate the pathogenicity of our rVHSVs, olive flounder (43.9 ± 7.43g) were i.p. injected with $2.3 \times 10^4$ PFU/fish or $2.3 \times 10^5$ PFU/fish of rVHSV-wild or rVHSV-P. Control fish were injected with phosphate buffered saline (PBS). On day 21 after viral infection, rVHSV-wild produced a cumulative mortality of 30% in fish infected with either $2.3 \times 10^4$ PFU/fish or $2.3 \times 10^5$ PFU/fish, whereas rVHSV-P showed cumulative mortalities of 55% and 80% in fish infected with $2.3 \times 10^4$ PFU/fish and $2.3 \times 10^5$ PFU/

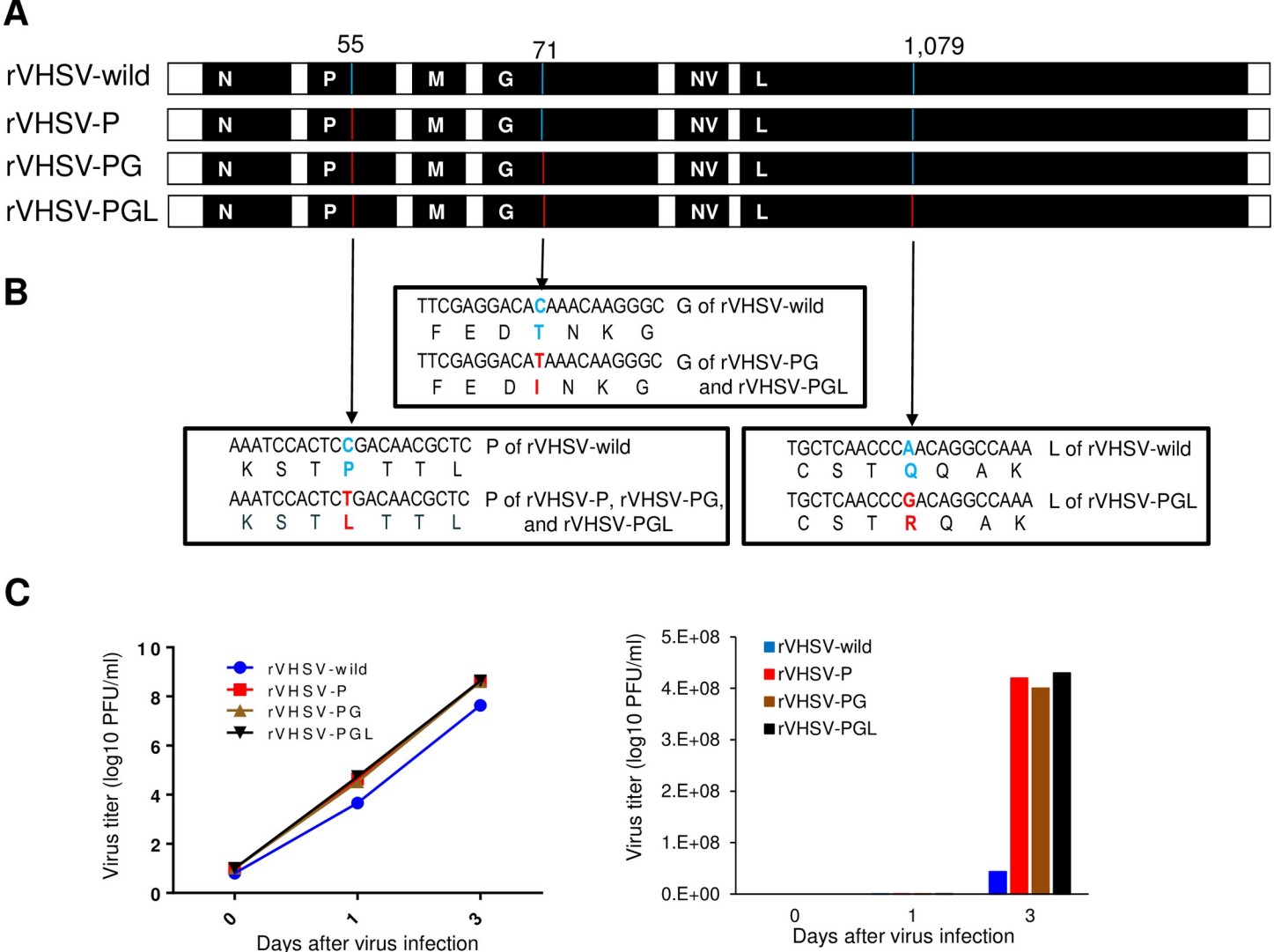

**Fig 2. Generation of recombinant VHSVs (rVHSVs) and their growth in HINAE cells.** (A) Schematic diagram of rVHSVs. rVHSV-P, one amino acid substitution, $P^{P55L}$; rVHSV-PG, two amino acid substitutions, $P^{P55L}$ and $G^{T71I}$; rVHSV-PGL, three amino acid substitutions, $P^{P55L}$, $G^{T71I}$, and $L^{Q1079R}$. (B) Nucleotide changes in the recombinant VHSVs were confirmed by nucleotide sequencing. Blue letters indicate the target nucleotide and amino acid sequences for site-directed mutation in rVHSV-wild, and red letters represent mutations in rVHSV-P, rVHSV-PG, and rVHSV-PGL. (C) Growth of rVHSVs in HINAE cells. The cells were infected with 0.01 MOI of rVHSVs at 14°C. At the indicated times, samples of the supernatant were collected and viral titers were determined by plaque assay. Left, logarithmic scale. Right, arithmetic scale.

fish, respectively (Fig 3). In the mock-infected negative control, mortality was 5%. These results indicate that the $P^{P55L}$ amino acid substitution enhances VHSV virulence in flounder.

## Effects of the $P^{P55L}$ amino acid substitution on viral genome replication and the transcription of VHSV

The P protein of a rhabdovirus acts as a cofactor of the viral RNA polymerase complex [32]. Therefore, we next tested whether the $P^{P55L}$ amino acid substitution affects viral RNA synthesis in HINAE cells. In VHSV-infected cells, viral RNA polymerase directs two types of RNA synthesis: mRNA synthesis (transcription) and genomic RNA amplification (replication). We first analyzed the viral mRNA and genomic RNA synthesis of a low-virulence ADC-VHS2015-

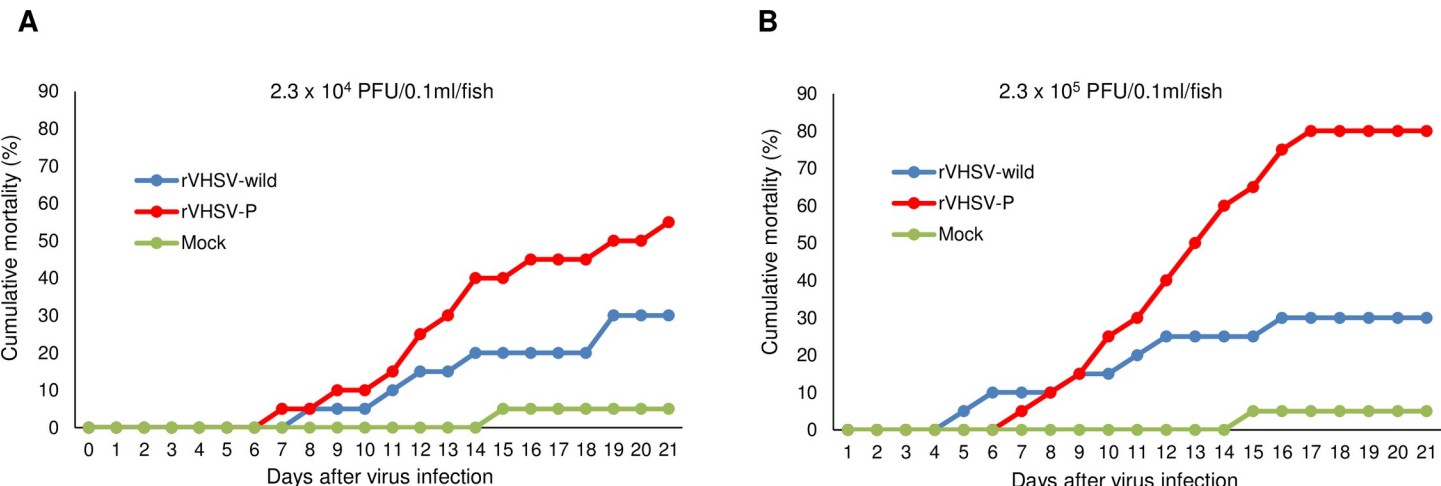

**Fig 3. Effect of the P^P55L amino acid substitution on VHSV virulence in flounder.** Graphs represent cumulative mortality of olive flounder infected with rVHSV-wild and rVHSV-P. Olive flounder (43.9 ± 7.43g) were intraperitoneally injected with (A) $2.3 \times 10^4$ PFU/fish or (B) $2.3 \times 10^5$ PFU/fish of rVHSV-wild or rVHSV-P at 13˚C. Fish injected with phosphate-buffered saline were used as the mock control. Fish mortality was monitored daily (n = 20 per group).

5 strain and a high-virulence ADC-VHS2012-6 strain in HINAE cells. The high-virulence ADC-VHS2012-6 strain expressed higher levels of both mRNA (+ strand) and genomic RNA (- strand) than the low-virulence ADC-VHS2015-5 strain (Fig 4A and 4B). To investigate the effect of P^P55L on viral mRNA and genomic RNA synthesis, HINAE cells were infected with the rVHSV-wild or rVHSV-P at an MOI of 1, and strand-specific quantitative RT-PCR assays were used to quantify the synthesis of viral mRNA and genomic RNA at 6, 12, and 24 h post-infection. At 12 h post-infection, rVHSV-P produced 12.4-fold higher levels of plus-sense viral RNA (transcription products and anti-genomes) (Fig 4C) and 15.4-fold higher levels of minus-sense RNA (genome copies) (Fig 4D) than rVHSV-wild. At 24 h post-infection, rVHSV-P produced 9.3-fold higher levels of plus-sense viral RNA (Fig 4C) and 8.6-fold higher levels of minus-sense RNA (Fig 4D) than rVHSV-wild. Taken together, these results indicate that P^P55L enhances the synthesis of viral mRNA and viral genomic RNA in cells.

## Ectopic expression of P^P55L enhances mRNA synthesis and the growth of low-virulence VHSV

To confirm whether P^P55L could enhance the RNA synthesis and growth of rVHSV-wild, we cloned the P gene from rVHSV-P to generate a plasmid, pcDNA6-P(P55L) plasmid. We also cloned the P gene from rVHSV-wild and generated a pcDNA6-P-wild. We transiently transfected HINAE cells with pcDNA6-P(P55L), pcDNA6-P-wild, or an empty pcDNA6 vector as a control (S2A Fig). The expression of P protein in each group of transfected cells was confirmed by western blotting (Figs 5A and S2B), and then the cells were infected with either the low-virulence ADC-VHS2015-5 strain or the high-virulence ADC-VHS2012-6 strain at an MOI of 1. At 24 h post-infection, we analyzed the RNA synthesis in the infected cells. Whereas ectopic expression of P^P55 decreased the (+) RNA synthesis of the high-virulence ADC-VHS2012-6 strain, ectopic expression of P^P55L increased the (+) RNA synthesis of both the low-virulence ADC-VHS2015-5 strain and the high-virulence ADC-VHS2012-6 strain (Fig 5B). Ectopic expression of neither P^P55 nor P^P55L affected (-) RNA synthesis of the low-virulence ADC-VHS2015-5 strain in HINAE cells (Fig 5C). However, (-) RNA synthesis of the high-virulence ADC-VHS2012-6 strain was decreased in HINAE cells overexpressing P^P55 but

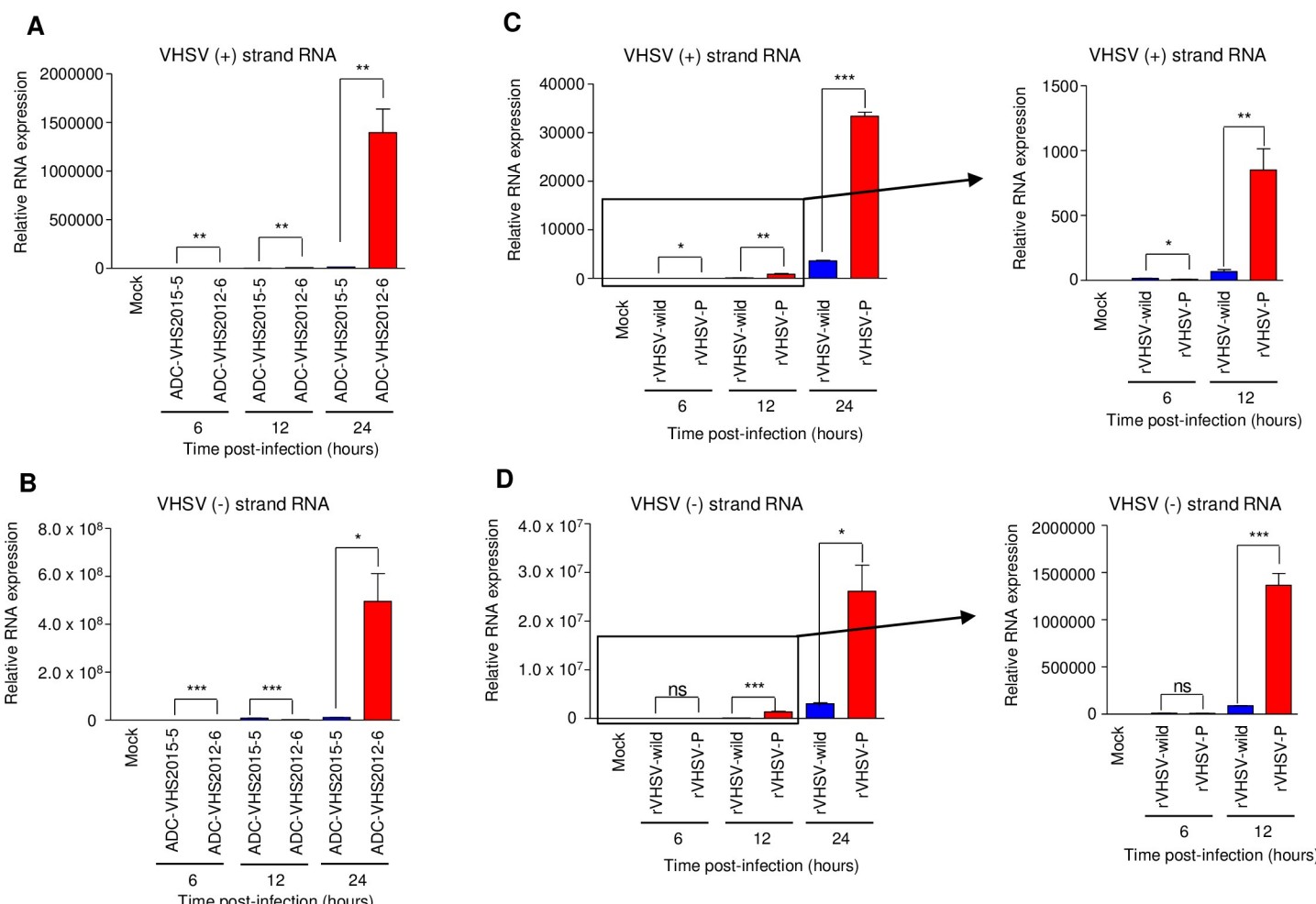

**Fig 4. Effect of the P$^{P55L}$ amino acid substitution on viral RNA synthesis in HINAE cells.** HINAE cells were infected with (A and B) low-virulence ADC-VHS2015-5, high-virulence ADC-VHS2012-6, (C and D) rVHSV-wild, or rVHSV-P at a multiplicity of infection of 1 PFU per cell, and cells were collected at the indicated time points. The accumulation of (B and D) VHSV genome copies (negative sense) and (A and C) G gene messenger RNA copies and anti-genomes (positive-sense) in the VHSV-infected cells was determined by strand-specific real-time PCR. The expression levels obtained from mock-infected cells were set to 1. The results are presented as the mean ± SD of three independent experiments. *, $p < 0.05$; **, $p < 0.01$; ***, $p < 0.001$; ns, not significant.

increased in cells overexpressing P$^{P55L}$ (Fig 5C). Thus, the P$^{P55L}$ amino acid substitution can enhance the viral RNA synthesis of VHSV in cells.

## Effects of the P$^{P55L}$ amino acid substitution on the expression of IFN genes *in vitro* and *in vivo*

The virulence and growth of many viruses depend on their ability to modulate the innate immune responses of infected cells [48–50]. The P protein of rabies virus antagonizes type I IFN-mediated antiviral responses by inhibiting the signaling pathways for IFN induction and response [33–37]. To determine whether P$^{P55L}$ amino acid substitution affects the IFN response in cells, HINAE cells were infected with 1 MOI of rVHSV-wild or rVHSV-P, total RNA was extracted at 6, 12, and 24 h post-infection, and the induction of IFN1 and Mx gene expression was analyzed by real-time PCR. Both rVHSV-wild and rVHSV-P induced the expression of IFN1, ISG15, and Mx in HINAE cells (Fig 6A–6C). At 6 h post-infection, the expression levels of IFN1 (Fig 6A), ISG15 (Fig 6B), and Mx (Fig 6C) in rVHSV-P-infected cells were lower

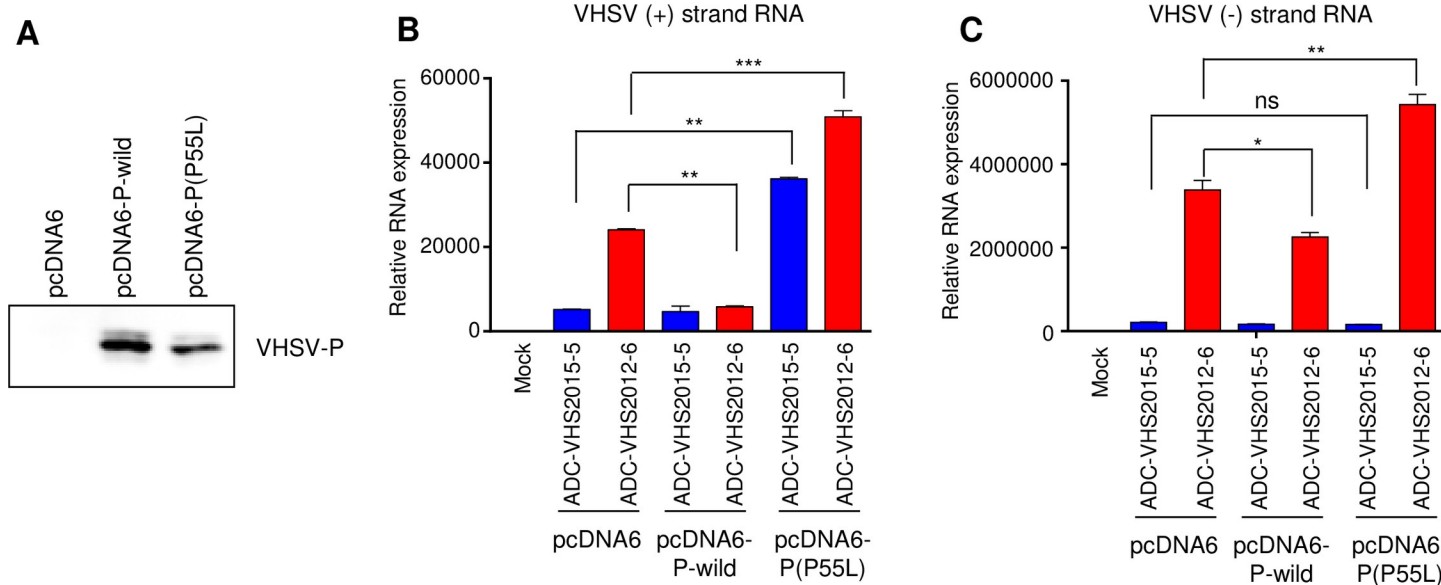

**Fig 5. Effects of the ectopic expression of VHSV P(P55) or VHSV P(P55L) on viral RNA synthesis.** HINAE cells were transfected with pcDNA6-P-wild or pcDNA6-P(P55L). An empty pcDNA6 vector was used as the control. Cells were treated with 5 μg/ml of blasticidin for 2 weeks to enrich the plasmid-bearing cells. (A) Western blot analysis of the expression of VHSV P in plasmid-transfected HINAE cells using an anti-V5 antibody. (B and C) Plasmid-transfected HINAE cells were infected with ADC-VHS2015-5 or ADC-VHS2012-6 at 1 MOI. At 24 h post-infection, cells were collected, and the accumulation of (B) G gene messenger RNA copies and anti-genomes (positive-sense) and (C) VHSV genome copies (negative sense) in the VHSV-infected cells was determined by strand-specific real-time PCR. The expression levels obtained from mock-infected cells were set to 1. The results are presented as the mean ± SD of three independent experiments. $^*$, $p < 0.05$; $^{**}$, $p < 0.01$; $^{***}$, $p < 0.001$; ns, not significant.

than those in rVHSV-wild-infected cells. However, at 12 h after infection, the expression of IFN1, ISG15, and Mx in rVHSV-P-infected cells increased to levels similar to those in rVHSV-wild-infected cells (Fig 6A–6C). By 24 h after infection, their expression in rVHSV-P-infected cells shifted toward lower levels than that in rVHSV-wild-infected cells (Fig 6A–6C).

We next i.p. injected olive flounder (43.9 ± 7.43g) with $2.3 \times 10^5$ PFU of rVHSV-wild or rVHSV-P and analyzed the expression levels of ISG15 and Mx in kidney, spleen, and liver samples collected at 0, 1 h, 6 h, 1 day, and 3 days after viral injection. Consistent with results obtained from the virus-infected HINAE cells, until 6 h after viral infection, the expression of ISG15 (Fig 6D) and Mx (Fig 6E) was down-regulated in the kidney, spleen, and liver of rVHSV-P-infected fish, whereas their expression was up-regulated in rVHSV-wild-infected fish. However, from 1 day after viral infection, the expression of the ISG15 and Mx genes in rVHSV-P-infected fish was up-regulated and, at 3 days after viral infection, their expression in rVHSV-P-infected fish increased to levels similar to those found in rVHSV-wild-infected fish (Fig 6D and 6E). These results suggest that P$^{P55L}$ inhibits the expression of IFN genes soon after VHSV infection, which could support the growth of VHSV. The increase in IFN gene expression at later time points after infection could be the result of increased viral growth.

## Ectopic expression of P$^{P55L}$ inhibits the expression of IFN genes in VHSV-infected cells

To confirm that supposition, HINAE cells were transiently transfected with pcDNA6-P$^{P55L}$, pcDNA6-P$^{P55}$, or empty pcDNA6/V5 vector as a control and infected with the low-virulence ADC-VHS2015-5 strain at an MOI of 1. At 6 h post-infection, the expression of IFN1 and Mx in the VHSV-infected cells was analyzed. Although ectopic expression of P$^{P55}$ did not affect the expression of either IFN1 or Mx, ectopic expression of P$^{P55L}$ decreased the expression of

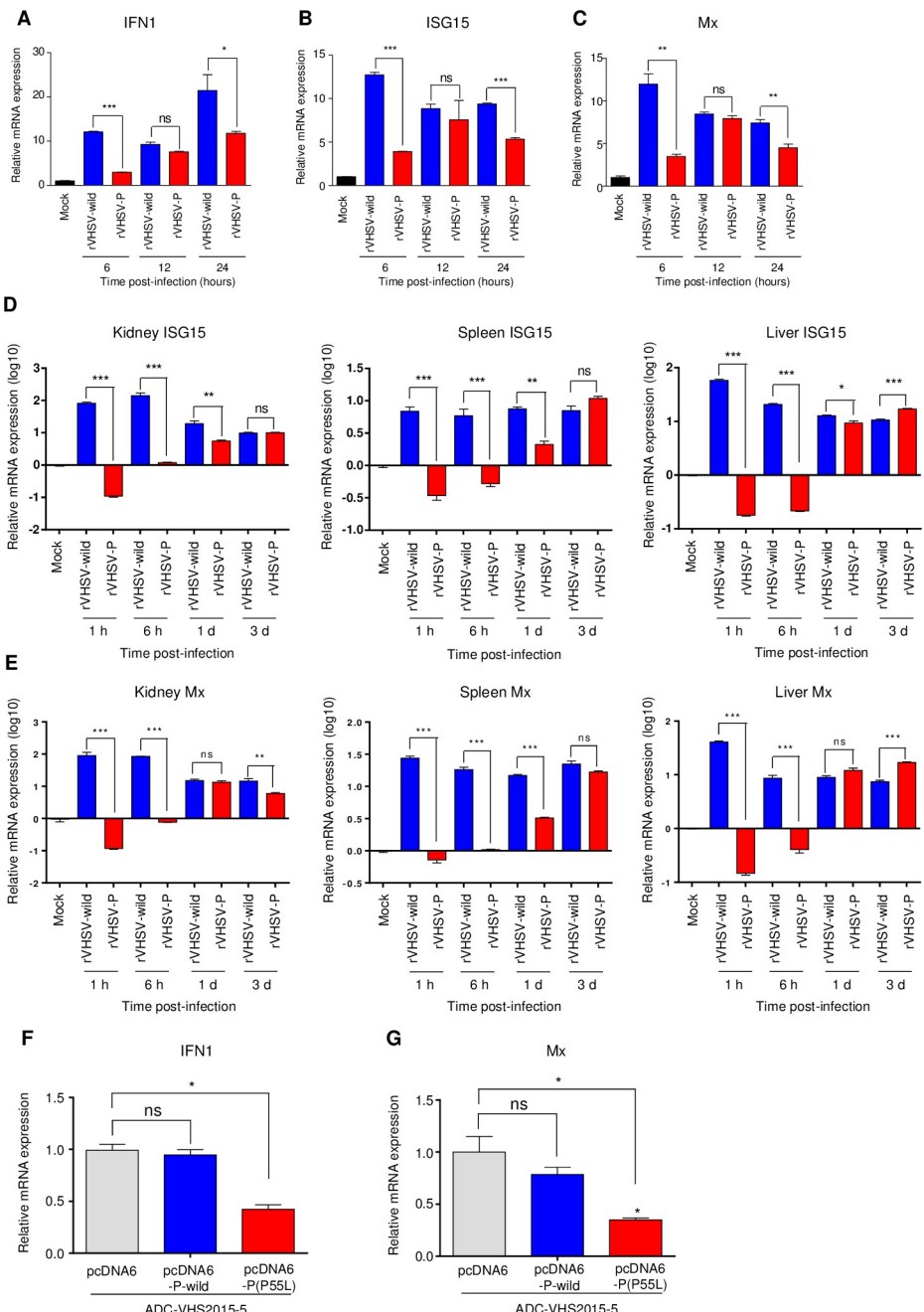

**Fig 6. Effects of the P^P55L amino acid substitution on the IFN response in HINAE cells and flounder.** (A–C) HINAE cells were infected with rVHSV-wild or rVHSV-P at 1 MOI, and cells were collected at the indicated time points. The expression levels of (A) IFN1, (B) ISG15, and (C) Mx in VHSV-infected HINAE cells were determined by real-time PCR. The expression levels obtained from mock-infected cells were set to 1. The results are presented as the mean ± SD of three independent experiments. *, $p < 0.05$; **, $p < 0.01$; ***, $p < 0.001$; ns, not significant. (D and E) Olive flounder (43.9 ± 7.43g) were intraperitoneally injected with $2.3 \times 10^5$ PFU/fish of rVHSV-wild or rVHSV-P at 13°C. At the indicated time points, kidney, spleen, and liver tissues were collected from anesthetized fish, and the expression levels of (D) ISG15 and (E) Mx in the tissues were determined by real-time PCR. The expression levels obtained from mock-infected fish were set to 1. The results are presented as the mean ± SD of three independent experiments. *, $p < 0.05$; **, $p < 0.01$; ***, $p < 0.001$. ns, not significant. (F and G) HINAE cells were transfected with pcDNA6-P-wild or pcDNA6-P(P55L). Empty pcDNA6 vector was used as the control. Cells were treated with 5 μg/ml of blasticidin for 2 weeks to enrich the plasmid-bearing cells. HINAE cells transfected with pcDNA6-P-wild, pcDNA6-P(P55L), or empty pcDNA6 vector as in Fig 5 were infected with ADC-VHS2015-5 at 1 MOI. At 6 h post-infection, cells were collected and the expression levels of (F) IFN1 and (G) Mx were determined by real-time PCR. The expression levels obtained

from mock-infected cells were set to 1. The results are presented as the mean ± SD of three independent experiments. *, $p<0.05$; ns, not significant.

both IFN1 (Fig 6F) and Mx (Fig 6G) in ADC-VHS2015-5-infected cells. These results suggest that the P$^{P55L}$ amino acid substitution can inhibit the expression of IFN1 and Mx in VHSV-infected cells.

## RNA-seq transcriptome analysis of rVHSV-infected HINAE cells

In this study we found that P$^{P55L}$ inhibits the expression of IFN genes at 6 h after VHSV infection (Fig 6). We next tested the effect of P$^{P55L}$ on the genome-wide response of flounder cells to VHSV infection at 6 h post-infection. To do this, we infected HINAE cells with 1 MOI of rVHSV-wild or rVHSV-P for 6 h and then explored the transcriptomes of those cells using RNA-seq. We detected 24,241 differentially expressed genes (DEGs) in the rVHSV-P-infected HINAE cells at 6 h post-infection, compared with rVHSV-wild-infected cells (S2 Table). Of them, we selected 396 DEGs down-regulated in rVHSV-P-infected HINAE cells compared with rVHSV-wild-infected HINAE cells and unambiguously mapped them to unique Entrez gene IDs. We conducted a GO enrichment analysis to identify the major gene groups down-regulated by the P$^{P55L}$ amino acid substitution. The GO enrichment analysis assigned the genes to 12 GO terms within the Biological Process category. The most prevalent GO terms were "Biological regulation" and "Response to stimulus" (Fig 7A).

Our aim in the DEGs analysis was to uncover the effect of P$^{P55L}$ on the expression of IFN-related genes in VHSV-infected HINAE cells. From the DEGs analysis, we identified 32 IFN-related genes (S3 Table). Of those 32 genes, 6 showed more than a 2-fold change in their expression in virus-infected cells at 6 h post-infection (Fig 7B and S3 Table). The rVHSV-wild-infected cells up-regulated 2 IFN-related genes (IRF10 and IRF1A) more than 2-fold compared to the mock-infected cells. However, rVHSV-P-infected cells did not up-regulate any IFN-related genes more than 2-fold compared to mock-infected cells (Fig 7B and S3 Table). Among the 6 genes with more than a 2-fold change in expression, 2 genes, IFITM5 and TRIM25L, were similarly down-regulated in both rVHSV-wild-infected and rVHSV-P-infected HINAE cells (Fig 7B and S3 Table), suggesting that P$^{P55L}$ might not affect the expression of those genes. The expression of the other 4 genes, IRF1A, IRF4A, IRF8, and IRF10, was down-regulated more than 2-fold in rVHSV-P-infected cells compared with rVHSV-wild-infected cells (Fig 7B and S3 Table). To further validate the results of RNA-seq, qRT-PCR was performed on IRF1A, IRF4A, IRF8, and IRF10 using HINAE cells infected with rVHSV-wild or rVHSV-P for 6 h. qRT-PCR analyses revealed the expression of those 4 genes was significantly reduced in rVHSV-P-infected cells compared with rVHSV-wild-infected cells (Fig 7C). The expression of those 4 genes might thus be suppressed by P$^{P55L}$ in the high-virulence ADC-VHS2012-6 strain.

## Discussion

With the emergence of virulent strains of VHSV that cause high mortality among cultured flounder in Asia [10–13], there is a growing need to better understand the genetic variations in VHSV strains that enhance virulence. In this study, we have investigated the genetic variations among 18 VHSV strains with different virulence levels in olive flounder and identified an amino acid substitution, P55L, in the P protein that correlates with VHSV virulence in flounder. Despite the inherent infidelity of the RNA polymerase of an RNA virus [51], this single amino acid polymorphism was stably maintained for 30 passages of naturally occurring variant

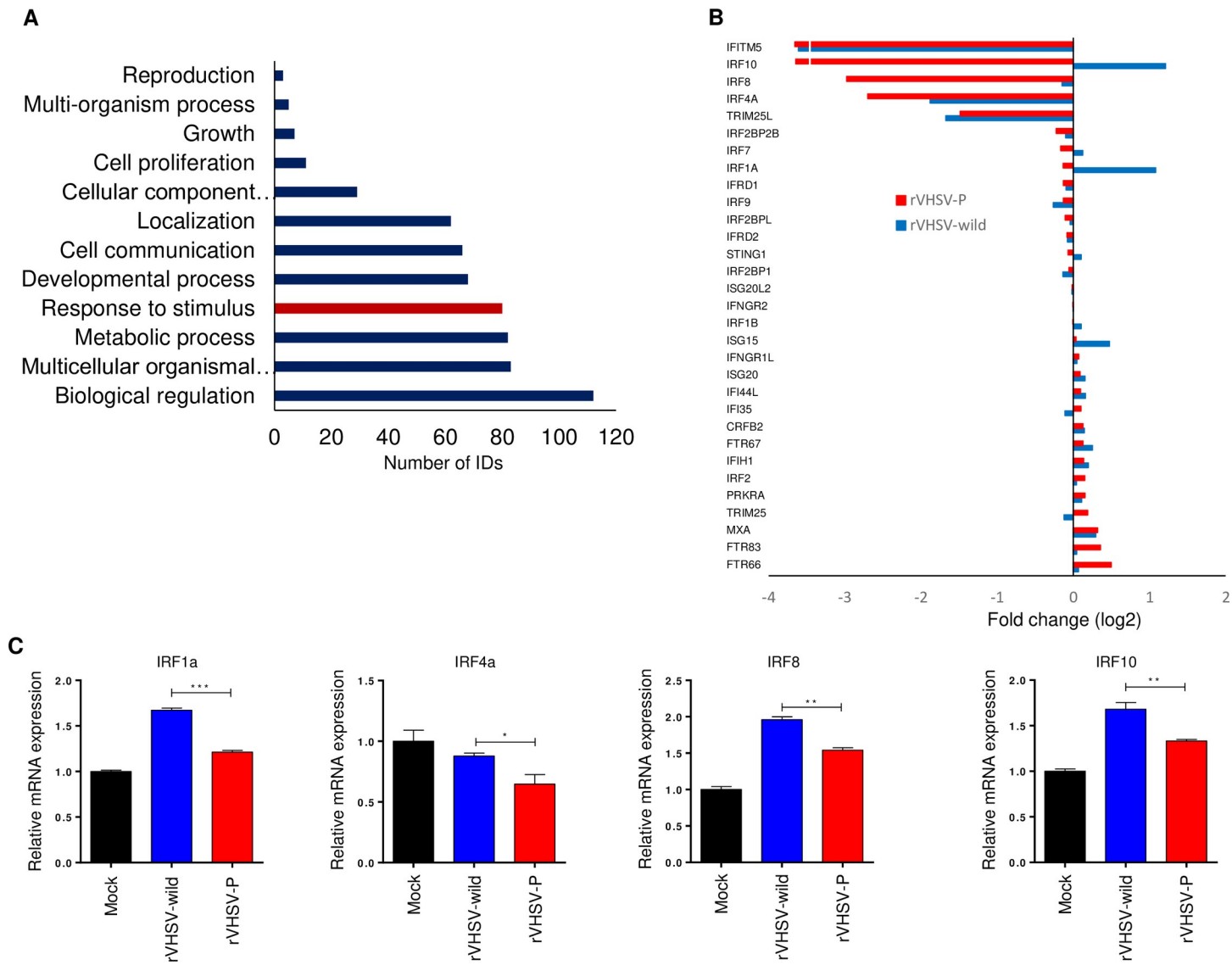

**Fig 7. Analysis of RNA-seq data from mock-, rVHSV-wild- and rVHSV-P-infected HINAE cells at 6 h post-infection.** (A) Unigenes down-regulated in rVHSV-P-infected HINAE cells compared to rVHSV-wild-infected HINAE cells were annotated to the 12 functional GO terms in the Biological Process category. (B) Fold changes in the expression of IFN response-related genes. The graph represents changes in the IFN response-related genes in rVHSV-wild- and rVHSV-P-infected cells compared to mock-infected cells. (C) HINAE cells were infected with rVHSV-wild or rVHSV-P at 1 MOI. At 6 h post-infection, cells were collected and the expression levels of IRF1A, IRF4A, IRF8, and IRF10 were determined by real-time PCR. The expression levels obtained from mock-infected cells were set to 1. The results are presented as the mean ± SD of three independent experiments. *, $p<0.05$; **, $p<0.01$; ***, $p<0.001$.

strains. Recombinant VHSV with $P^{P55L}$ (rVHSV-P) showed significantly enhanced viral virulence *in vitro* and *in vivo* compared with rVHSV containing $P^{P55}$ (rVHSV-wild).

Type I IFNs are induced by VHSV infection [40] and inhibit VHSV growth in flounder [41, 42]. It is generally accepted that the virulence and growth of many viruses depend on their ability to suppress innate immune responses in virus-infected cells [48–50, 52]. Rhabdoviruses rely on the P protein to inhibit signaling pathways for IFN induction and response [33–37]. Thus, it is possible that $P^{P55L}$ enhances the virulence of VHSV by increasing its ability to block IFN induction and response. In this study, we provide evidence supporting this hypothesis. At 6 h post-infection, rVHSV-P blocks the induction of IFN1, ISG15, and Mx relative to rVHSV-

wild in both HINAE cells and flounder tissues. In addition, a GO enrichment analysis of RNA-seq data demonstrated a decrease in "Response to stimulus" in the "Biological process" category in rVHSV-P-infected cells at 6 h post-infection compared with rVHSV-wild-infected cells. Specifically, rVHSV-P infection inhibited the expression of the IRF10, IRF8, IRF4A, and IRF1A IFN-related genes compared with rVHSV-wild infection. Considering the antiviral function of the IFN response [53, 54], our results suggest that VHSV containing the $P^{P55L}$ amino acid substitution blocks the induction of the IFN response more efficiently than VHSV with $P^{P55}$ during the early stage of virus infection, which could facilitate the growth of VHSV and thus enhance VHSV virulence in both cells and fish. Consistently, we found that ectopic expression of $P^{P55L}$ blocked the induction of the IFN response in cells infected with a low-virulence VHSV strain at 6 h post-infection. However, at 12 h and 3 days post-infection in rVHSV-P-infected cells and fish, respectively, the expression of ISG15 and Mx was similar to that in rVHSV-wild-infected cells and fish. It has been reported that rapidly growing, high-virulence fish rhabdovirus induces a higher IFN response than slow-growing, low-virulence virus, but the IFN response induced by rapidly growing viruses does not correlate with protection [55–58]. Consistently, we previously reported that although poly I:C treatment before virus infection significantly inhibited the growth of fish rhabdovirus in cells, poly I:C treatment after virus infection did not [45]. The increase in IFN1 and Mx expression in rVHSV-P-infected cells and fish at 24 h post-infection could thus be the result of rapid viral growth and might not be critical in inhibiting viral growth. These results suggest that the $P^{P55L}$ amino acid substitution could support VHSV growth by blocking the host IFN response at the early stage of virus infection.

In addition to blocking the host's innate immune response, the rhabdovirus P protein supports viral growth by acting as a cofactor of viral RNA polymerase [32]. We here found that the expression of viral mRNA and genomic RNA in HINAE cells infected with rVHSV-P was significantly higher than in those infected with rVHSV-wild. More important, we found that ectopic expression of $P^{P55L}$ enhanced the expression of rVHSV-wild mRNA, and ectopic expression of $P^{P55}$ reduced the expression of rVHSV-P mRNA in cells. Thus, the $P^{P55L}$ amino acid substitution enhances the transcription of VHSV. It has been reported that rhabdovirus counteracts the host antiviral immune responses by using multiple viral proteins. Rabies virus N [59] and P [33] proteins suppress the host IFN response by inhibiting RIG-1 activation and blocking IRF3 phosphorylation, respectively. Rabies virus M proteins suppresses NF-kB activity and the expression of antiviral cytokines [60]. In addition, VHSV NV inhibits IFN response by reducing TBK1 phosphorylation [61]. Even though we did not analyze the effect of the $P^{P55L}$ amino acid substitution on the expression of viral proteins, it is possible that enhanced transcription of VHSV by the $P^{P55L}$ amino acid substitution may lead to increased expression of VHSV proteins, which plays roles in counteracting the host antiviral immune responses. Until now, the regulation mechanisms of VHSV RNA synthesis have not been well studied. However, work with the mammalian VSV rhabdovirus has revealed that viral RNA synthesis is carried out by two RNA polymerase complexes: a complex of the viral L and P proteins for transcription and a complex of the viral L, P, and N proteins for replication [62, 63]. Both viral transcription and replication are affected by phosphorylation of the viral P protein by host kinases [64, 65]. By using NetPhosK [66], we were able to predict kinase-specific phosphorylation sites within the VHSV P protein. The $P^{P55L}$ amino acid substitution did not change the number or position of the phosphorylation sites within the P protein. However, interestingly, the $P^{P55L}$ amino acid substitution did induce a change in kinase-specificity at the threonine 54 residue (T54). Whereas $P^{P55}$ contains a p38MAPK phosphorylation site at T54, $P^{P55L}$ has a PKA phosphorylation site at T54 (S3A Fig). Because there is no commercially available antibody recognizing a VHSV P protein phosphorylated at T54 and no inhibitors specific to

flounder p38MAPK and PKA, we could not confirm whether $P^{P55L}$ and $P^{P55}$ are differentially phosphorylated by p38MAPK and PKA, respectively, in VHSV-infected flounder cells or whether this phosphorylation is essential in enhancing viral RNA synthesis and inhibiting the IFN response. However, we hypothesize that phosphorylation of the VHSV P protein might be required for efficient viral RNA synthesis and possibly for blocking the host IFN response. The $P^{P55L}$ amino acid substitution could change kinase-specificity at T54 from p38MAPK to PKA, which would lead to enhanced phosphorylation of the P protein in VHSV-infected HINAE cells (S3B Fig).

In this study, we have identified the $P^{P55L}$ amino acid substitution in the VHSV P protein and shown its correlation with VHSV virulence in flounder. Recombinant VHSV containing $P^{P55L}$ had enhanced virulence in flounder, enhanced viral RNA synthesis, and blocked the host IFN response better than rVHSV containing $P^{P55}$. Considering that rhabdoviruses rely on their P protein to suppress the IFN response [33–37] and enhance RNA synthesis [32, 64, 67], our results suggest that the higher virulence of VHSVs containing $P^{L55}$ is caused by enhanced suppression of the host IFN response and enhanced viral RNA synthesis in cells. The identification of the $P^{P55L}$ amino acid substitution as a virulence determinant could contribute to our understanding how VHSV mutation determines the VHSV virulence in flounder. It could also facilitate the prediction and control of VHSV epizootics.

## Materials and methods

### Ethics statement

All fish were handled in accordance with the Guide for the Care and Use of Laboratory Animals from the National Research Council. All animal procedures were approved by the Institutional Animal Care and Use Committee (IACUC) of the National Institute Fisheries Science (permit number 2017-NIFS-IACUC-04). All efforts were made to minimize suffering by sacrificing fish under tricaine mesilate (MS-222) anesthesia.

### Cell line

Hirame natural embryo (HINAE) cells were grown at 20˚C in Leibovitz medium (Welgene) supplemented with 10% fetal bovine serum and epithelioma papulosum cyprini (EPC) cells were grown at 20˚C in Eagle's minimal essential medium (EMEM) (Welgene) supplemented with 10% fetal bovine serum (Welgene). For the amplification of recombinant VHSV, EPC cells were cultured in EMEM supplemented with penicillin (100 U ml$^{-1}$), streptomycin (100 µg ml$^{-1}$) and 10% fetal bovine serum (Welgene).

### Viruses

Eighteen VHSV strains isolated from olive flounder cultured in Korea were used in this study. The source, place of location, and year of isolation of the VHSV strains used in this study are summarized in S1 Table [43]. Viruses were propagated in EPC cells at 14˚C and quantified in terms of PFU/ml using a standard plaque assay [68].

### Pathogenicity experiments in flounder

To test the pathogenicity of the VHSV strains, olive flounder were experimentally infected. Olive flounder were purchased from a commercial fish farm without a history of VHS and 20 fish were stocked in each tank and acclimated at 11–13˚C for 1 week. The 18 VHSV strains and two recombinant VHSVs used in the pathogenicity test were propagated and titrated on to EPC cells using a plaque assay. Four independent experiments were conducted to determine

the pathogenicity of 18 VHSV strains using different lots of fish: experiment 1, 29.61 ± 5.55 g; experiment 2, 31.85 ± 3.89 g; experiment 3 and 4, 39.7 ± 14.63 g (mean ± SD). The fish in the tanks were intraperitoneally (i.p.) injected with the same dose of $1 \times 10^4$ PFU per fish or 100 μl of control medium at 11–13°C. Two independent experiments were conducted to determine the pathogenicity of 2 recombinant VHSVs using the same lot of fish (43.9 ± 7.43g). The fish were i.p. injected with different dose of recombinant viruses: experiment 1, $2.3 \times 10^4$ PFU per fish; experiment 2, $2.3 \times 10^5$ PFU per fish. Tanks were monitored twice daily for fish showing signs of ill health. Clinical signs of the disease and mortality were recorded daily for 14 days. Virus was re-isolated from kidney and spleen homogenates of moribund and dead fish by using EPC cells.

## Virus growth curve analysis

Infections were conducted in 25-cm$^2$ culture flasks containing confluent monolayers of HINAE cells at an MOI of 0.1 PFU per cell. At the end of a 30 min absorption period at 20°C, the inoculum was removed, and the cells were washed three times with MEM containing no serum. MEM (5 ml) containing 5% fetal bovine serum was then added to each culture flask and the flasks were incubated at 20°C. Samples, consisting of 200 μl of medium, were taken at 0, 24, 72, and 120 h post-infection and stored at –80°C. Viral titer was determined for all samples, in duplicate, by a plaque assay using EPC cells. Growth curves were constructed using the mean log titer for each time point.

## RNA extraction

HINAE cells were harvested 12 and 24 h after infection with 1 MOI of VHSV. Mock-infected HINAE cells were used as controls. Head kidney, spleen, and liver tissues were excised from flounder at 1 h, 6 h, 1 days, and 3 days after intraperitoneal injection with a dose of $2.3 \times 10^5$ PFU VHSV per fish. Tissues from mock-infected flounder were used as the control. Total RNA was isolated from cells and tissues using TRIzol reagent (Invitrogen).

## Quantification of gene expression by qRT-PCR

Two μg of total RNA was used to generate cDNA with M-MLV reverse transcriptase (Promega). Real-time qRT-PCR was performed using SYBR Green PCR Master Mix (Qiagen, Hilden, Germany) on an ABI 7500 Fast Real-Time PCR System (Applied Biosystems). The primer pairs are listed in S4 Table. Following an initial 10 min denaturation/activation step at 95°C, the mixture was subjected to 40 cycles of amplification (denaturation for 15 s at 95°C, annealing and extension for 1 min at 60°C). The specificity of the qPCR reaction for each amplified product was verified by melting curve analyses.

## Quantification of viral transcription and viral genomic RNA by qRT-PCR

Viral genomic RNA (negative-strand RNA) and mRNA (positive-strand RNA) were determined by strand-specific qRT-PCR assays. Total RNA was isolated from virus-infected cells using the TRIzol extraction method (Invitrogen), and cDNA was synthesized using 2 μg of total RNA and strand-specific primers (positive-strand-specific primers for mRNA, 5' GGC AGT ATC GTG AAT TCG ATG CGG TGT CCC CAT GAG TTC GAG 3'; negative-strand-specific primers for genomic RNA, 5' GGC AGT ATC GTG AAT TCG ATG CAC TGC TGA GAC GCT GGT GAC 3'; Tag primer, 5' GGC AGT ATC GTG AAT TCG ATG C 3'; VHSV G-AS, 5' GGT GTC CCC ATG AGT TCG AG 3'; VHSV G-S, 5' ACT GCT GAG ACG CTG GTG AC 3'). Quantitative RT-PCR was performed as described above.

## Generation of recombinant viruses

Three recombinant VHSVs were generated using plasmid-based reverse genetics as previously described [44]. Sequential mutations P(P55L), G(T71I), and L(Q1,079R) were introduced into the pVHSV-wild plasmid [44] using specific oligonucleotide primers (S5 Table) and a Quik-Change Site-Directed Mutagenesis Kit (Stratagene, La Jolla, CA) according to the manufacturer's instructions. Following an initial 10 min pre-denaturation step at 95˚C, the mixture was subjected to 12 cycles of amplification (denaturation for 30 s at 95˚C, annealing for 1 min at 55˚C, and extension for 14 min at 68˚C) and then final extension for 7 min at 72˚C. After incubation with *Dpn*I for 2 h at 37˚C, the PCR products were introduced into *Escherichia coli* DH5α, and plasmid DNA was extracted using an Exprep Plasmid SV mini (GeneAll). EPC cells stably expressing T7 RNAP were grown to about 80% confluence and co-transfected with a mixture of pVHSV-wild or pVHSV-P(P55L) and 3 plasmids encoding VHSV N (pCMV-N, 500 ng), P (pCMV-P, 300 ng), and L (pCMV-L, 200 ng) using FuGENE 6 (Roche) according to the manufacturer's instructions. Transfected cells were incubated for 12 h at 28˚C, and then further incubated at 15˚C. When total cytopathic effect was observed, the cells were suspended by scraping the plates with a rubber policeman. Then they were subjected to 2 cycles of freeze-thawing and centrifuged at $4000 \times g$ for 10 min. The rVHSVs in the supernatant (named P0) was used to inoculate fresh EPC cell monolayers in a T25 flask at 15˚C. At 7 to 10 d post-inoculation, the supernatant (P1) was harvested, aliquoted, and stored at −80˚C. To confirm the point mutations in the P, G, and L genes of the rVHSVs, total RNA was extracted from the P1 supernatant using RiboEX(GeneAll) and a Hybrid-R kit (GeneAll). After cDNA synthesis using a cDNA synthesis kit (Promega), the P, G, and L genes were PCR amplified using Hipi taq and PCR primers (S3 Table). Following an initial 3 min pre-denaturation step at 95˚C, the mixture was subjected to 30 cycles of amplification (denaturation for 30 s at 95˚C, annealing for 1 min at 60˚C, and extension for 1 min at 72˚C) and then final extension for 7 min at 72˚C. The PCR products were ligated to pGEM-T vectors (Promega) and subjected to nucleotide sequencing.

## Cloning and expression of VHSV P protein in HINAE cells

Full-length cDNA of the P genes of ADC-VHS2015-5 (containing P55 in the P protein) and ADC-VHS2012-6 (with the P55L amino acid substitution in the P protein) were amplified from the cDNA of virus-infected HINAE cells using PCR and PCR primers (S4 Table). The PCR products were ligated into the *Hind*III/*Eco*RI sites of pcDNA6/V5-His(Invitrogen) mammalian expression vectors to generate the plasmids pcDNA6-P-wild and pcDNA6-P(P55L). HINAE cells were transfected with those plasmid vectors using Lipofectamine 3000 (Thermo Fisher Scientific, L3000015) and treated with blasticidin for 1 week to enrich them. The expression of the P genes in HINAE cells was confirmed by a western blot analysis using an anti-V5 monoclonal antibody (GenWay Biotech Inc.). VHSV P gene–transfected cells were infected with 1 MOI of ADC-VHS2015-5 or ADC-VHS2012-6. At the indicated times after viral infection, the cells were analyzed for their expression of IFN genes and viral RNA using quantitative real-time PCR.

## RNA preparation and RNA-seq

Total RNA was isolated from rVHSV-HINAE and rVHSV-P cells using an RNeasy mini kit (Qiagen). The 28S/18S ratio of the total RNA in each sample was determined using a Nano-Drop 2000 system (Thermo Fisher Scientific, Waltham, MA, USA). The cDNA library was prepared with ∼1.0 μg of total RNA, following the manufacturer's recommendations from a TrueSeq RNA Library Preparation Kit (Illumina, San Diego, CA, USA). cDNA was amplified according to the RNA-seq protocol provided by Illumina and sequenced using an Illumina

HiSeq 2500 system to obtain 150-bp paired-end reads. The sequencing depth for each sample was >20 million reads. The Trinity (r2013-11-10) pipeline program was used to assess and assemble contigs. Transcripts assembled from the total reads in each mRNA sample were merged and mapped to the zebrafish genome, together with the olive flounder genome, using Tophat v2.0.4 with default parameters. Only those reads aligned against zebrafish or flounder genomes were considered in the DEG analysis with Cuffdiff [69].

## DEGs and enriched GO and pathway analyses

Transcript abundances, in reads per kilobase per million reads mapped, were estimated using RNA-seq with expectation maximization (RSEM) through the Trinity plug-in, run_RSEM.pl. DEGs between the mock-infected control and VHSV-infected cells were identified using the DEseq package in R software. To identify the differential expression patterns of transcripts the TMM-normalized FPKM matrix (FPKM = total exon fragments/[mapped reads (millions)×exon length (kb)]) was used to generate heat maps in an R programming environment. Functional annotations were conducted by comparing sequences with public databases. All Illumina-assembled unigenes were compared with the NCBI non-redundant protein database (http://www.ncbi.nlm.nih.gov/) and the Kyoto Encyclopedia of Genes and Genomes (KEGG) database (http://www.genome.jp/kegg) using NCBI BLAST (http://www.ncbi.nlm.nih.gov/). Gene Ontology (GO) terms were assigned to each unigene based on the GO terms annotated to its corresponding homologs. Unigenes were classified according to GO terms within molecular functions, cellular components, and biological processes. Unigenes were assigned to special biochemical pathways according to the KEGG database using BLASTx, followed by retrieving KEGG Orthology information.

## Statistical analysis

Differences in the expression of innate immune response genes and differences in the growth of the two VHSV strains were evaluated by Student's t-test or one-way ANOVA. Cumulative mortalities at the end of the trial within and between different groups in each experiment were compared by chi-square analysis. A P value less than 0.05 was considered statistically significant.

## Supporting information

**S1 Fig. Comparison of the nucleotide sequences of the P gene between low- and high-virulence VHSV strains after serial passaging.** Low-virulence (ADC-VHS2015-5) and high-virulence (ADC-VHS2012-6) VHSV strains were continuously sub-cultured in HINAE cells for 30 passages. The VHSV P genes in passages 15 (P15), 20 (P20), 25 (P25), and 30 (P30) were amplified by PCR, and each base in the PCR products was sequenced an average of three or four times. The three bases in the boxed region represent the codon for the 55th amino acid residue of VHSV P. The blue bases are from ADC-VHS2015-5, and the red bases are from ADC-VHS2012-6.
(TIF)

**S2 Fig. Cloning and expression of the VHSV P gene.** Full-length open reading frame of P genes from rVHSV-wild and rVHSV-P were PCR amplified and cloned into the *Hind*III/*EcoR*I sites of the mammalian expression vector pcDNA6/V5-His A to generate pcDNA6-P-wild and pcDNA6-P(P55L), respectively. (A) Agarose gel electrophoresis image of VHSV P gene inserts in the plasmids after cutting them with *Hind*III/*EcoR*I. (B) Each plasmid was transiently transfected into HINAE cells for 24 h, and the cell lysates were analyzed for the VHSV

P protein by immunoprecipitation using an anti-V5 antibody.
(TIF)

**S3 Fig. Proposed role of the P^P55L amino acid substitution in the regulation of viral RNA synthesis and host IFN response.** (A) Prediction of the kinase-specific phosphorylation site at T54 of the VHSV P(P55) and P(P55L) protein using the NetPhosK server. (B) Proposed model for the role of the P^P55L amino acid substitution in regulating viral RNA synthesis and host IFN response. The P^P55L amino acid substitution could change the kinase-specificity at T54 from p38MAPK to PKA, leading to enhanced phosphorylation of the P protein in VHSV-infected HINAE cells, which would increase the RNA polymerase activity of the VHSV L-N-P complex and block the host IFN response.
(TIF)

**S1 Table. VHSV strains used in this study.**
(DOCX)

**S2 Table. DEGs in rVHSV-infected HINAE cells (rVHSV-wild vs rVHSV-P).**
(XLSX)

**S3 Table. IFN-related DEGs (Mock vs rVHSV-wild or rVHSV-P).**
(XLSX)

**S4 Table. PCR primers used in this study.**
(DOCX)

**S5 Table. PCR primers used to generate the recombinant VHSVs.**
(DOCX)

## Author Contributions

**Conceptualization:** Ki Hong Kim, Chan-Il Park, Jeong Woo Park.

**Data curation:** Jee Youn Hwang, Unn Hwa Lee, Min Jin Heo, Min Sun Kim, Bo Young Jee, Ki Hong Kim, Chan-Il Park, Jeong Woo Park.

**Formal analysis:** Jee Youn Hwang, Unn Hwa Lee, Min Jin Heo, Min Sun Kim, Ji Min Jeong, So Yeon Kim, Mun Gyeong Kwon.

**Funding acquisition:** Jeong Woo Park.

**Investigation:** Unn Hwa Lee, Min Jin Heo, Ji Min Jeong, So Yeon Kim, Ki Hong Kim, Chan-Il Park, Jeong Woo Park.

**Methodology:** Jee Youn Hwang, Unn Hwa Lee, Min Sun Kim.

**Supervision:** Jeong Woo Park.

**Validation:** Mun Gyeong Kwon, Bo Young Jee, Ki Hong Kim, Chan-Il Park.

**Writing – original draft:** Jeong Woo Park.

**Writing – review & editing:** Ki Hong Kim, Chan-Il Park, Jeong Woo Park.

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
