## [Decision Letter · Decision Letter 0]

11 Sep 2020

Dear Dr. Park,

Thank you very much for submitting your manuscript "Naturally occurring substitution in one amino acid in VHSV phosphoprotein enhances viral virulence in flounder" for consideration at PLOS Pathogens. Firstly, we thank you for your patience, as the pandemic has taken up a lot of the editors' and reviewers' time. As with all papers reviewed by the journal, your manuscript was first reviewed by members of the editorial board. This section editor thought it was important to include pathogenesis studies on non-human/non-mammalian pathogens as long as the studies were well-done and inform the field on general pathogenic principles of a given pathogen class.  Your studies on VSHV, an important viral pathogen in the aquaculture industry, appears to fulfill that purpose. Furthermore, the virulence determinants studied in VSHV may allow for comparative pathogenesis studies amongst the Rhadoviridae, one of the largest families amongst the mononegavirads. 

To ensure that your manuscript received due consideration, we managed to recruit two world experts in the subject matter who provided very constructive reviews. The reviewers appreciated the attention to an important topic. Based on the reviews, we are likely to accept this manuscript for publication, providing that you modify the manuscript according to the review recommendations.

As you can see, both reviewers though the mechanistic work was well-performed and informative. Reviewer #2 had only a list of minor comments which can (and should) be addressed by textual revisions. Reviewer #1 also thought the experiments well-designed and logicaly presented, but this reviewer suggested that there might be alternative explanations for attributing the designated P mutations to anti-host interactions. In this regard, reviewer#1 suggested that a time-series western blot of VSHV proteins might help differentiate between alternate interpretations of your data. My recommendation is to do the experiment if this is easily done with the reagents at hand. However, given the vagaries of how this COVID-19 pandemic has unfolded, I would also accept a revision that addresses Reviewer #1's concerns textually (e.g. by acknowledging the alternate interpretations or providing other pieces of supopirting data).   

As indicated above, if your revisions explicitly address the Reviewers' (mostly) minor concerns, we are prepared to accept your revised manuscript without sending it out for further review. 

Sincerely,

Benhur Lee

Section Editor

PLOS Pathogens

Benhur Lee

Section Editor

PLOS Pathogens

Kasturi Haldar

Editor-in-Chief

PLOS Pathogens

orcid.org/0000-0001-5065-158X

Michael Malim

Editor-in-Chief

PLOS Pathogens

orcid.org/0000-0002-7699-2064

Reviewer Comments (if any, and for reference):

Reviewer's Responses to Questions

**Part I - Summary**

Reviewer #1: The manuscript by Hwang et al, “Naturally occurring substitution in one amino acid in VHSV phosphoprotein enhances viral virulence in flounder”, describes a series of studies aimed at determining whether VHSV IVa strains, isolated from flounder during outbreaks in 2012-2016, harbored mutations that contributed to the variations in virulence associated with these isolates. Whole viral genome sequencing, coupled with mortality studies on all 18 strains, allowed the authors to separate the strains into high virulence (mortality), and low virulence strains. Among the observed sequence variations, one in particular correlated with high virulence. This P-to-L substitution at amino acid 55 of the Phosphoprotein (P) was observe in 11/12 of the highest virulence strains, but none of the six lowest virulence strains. To assess the significance of this correlation, the authors conducted a series of studies testing the effect of this mutation, alone and in combination with other changes, in recombinant VHSV mutants. The studies showed that the P(P55L) mutation led to enhanced replication, mortality and host immune gene suppression.

The data presented provide an interesting and fairly comprehensive story about a potential role for P(P55L) in regulating VHSV virulence in flounder. This was a logical, well designed and well written paper I had only a few minor comments about the studies.

Reviewer #2: This manuscript describes very thorough and sophisticated studies of the genetic and molecular basis of virulence for the fish rhabdovirus VHSV. VHSV is a globally significant pathogen of several major aquaculture fish species, and has a direct impact on both fisheries economics and natural ecosystem health. In this paper the virulence of VHSV in the predominant aquaculture fish in Asia, olive flounder, follows excellent scientific logic and experimental designs. The results are presented succinctly and clearly, and conclusions are well supported and significant. The writing is well organized and smooth for reading. The experimental studies conducted comprise an impressive array of work starting with replicated in vivo fish infection trials to quantify virulence and identify multiple high and low virulence strains, followed by identification of amino acid differences based on whole genome sequences. A total of three amino acids were found to correlate with the virulence phenotype, and all three were investigated by in vitro mutagenesis of an existing infectious clone. Recombinant viruses were thoroughly characterized to demonstrate that a single amino acid substitution in the VHSV phosphoprotein, PP55L, results in increased replication in vitro and increased virulence in vivo. Further, numerous molecular assays demonstrated that the PP55L mutation impacts both plus- and minus viral RNA synthesis, and blocks interferon responses as assessed by individual gene qPCR and transcriptomics. The use of strand-specific qPCR assays and ectopic expression of the relevant P protein variants are novel aspects of the work that contribute to the well-supported body of evidence providing a full understanding of the impact of this mutation. Overall I find this work to be novel and broadly of interest in presenting a new element the understanding of rhabdovirus virulence mechanisms. My suggestions for the authors to consider in making the paper stronger are detailed below, but they all deal with relatively minor aspects that can be easily corrected.

**Part II – Major Issues: Key Experiments Required for Acceptance**

Reviewer #1: Although the work was a bit phenomenological with respect to the impact of the mutation on P function, the authors focused on the impact on IFNs and ISG expression. The data in figures 6 and 7 showed that at certain timepoints, the impact of the mutant virus (or protein) impacted innate immune gene induction. The authors hypothesized that the mutant P protein suppressed innate immune gene induction more effectively, thus allowing more viral replication at early stages, giving the virus a head start on growth as compared to the control virus and allowing more rapid growth and, ultimately, mortality.

While this is perfectly fine as a hypothesis, the authors spend a bit too much effort attributing this to the anti-host function of P. But the data showing a dramatic impact of this mutant on viral RNA synthesis, particularly + strand, suggests that it would potentially alter all viral proteins, any one of which could be impacting host function (multiple have been implicated in this role). In other words, the impact on viral gene transcription could be the primary impact of the mutation, with the subsequent effects indirectly mediated by the other viral proteins/genes, especially if the mutation alters the normal balance or kinetics of anti-host gene induction.

A simple but important experiment to share with the reader along these lines is a Western Blot showing the accumulation of viral proteins post-infection, beginning at early timepoints. This could be done in multiple ways in multiple cells systems that the authors have developed, and would provide supporting data one way or another to direct future studies. It is appreciated that the authors feel that rapid growth alone cannot alter host suppression, but altered anti-host gene expression kinetics could. A pan-VHSV antibody that detects all proteins would allow this experiment to be conducted rapidly, and perhaps expand the possible interpretations, or support the hypothesis proposed by the authors.

Reviewer #2: no further experiments are required.

**Part III – Minor Issues: Editorial and Data Presentation Modifications**

Reviewer #1: Figure 3 - please label the viruses used in the studies the same in the graph inset

Within the text that discusses “Effects of the substitution on the expression of IFN genes in vitro and in vivo” the authors claim that at 12 and 24hpi the expression of IFN and Mx in P55L infected cells increased to levels similar to WT infected cells. This is not supported by the data in figure 6a where the 24 hpi data are not very similar to wt (although they are at 12 hpi). The narrative simply needs to align witht he data better.

Reviewer #2: 1. Results line 1 say the 18 strains were isolated from olive flounder during "an outbreak" of VHS in Korea between 2012-2016. The words "an outbreak" suggest they were all from the same location, which is not a very broad sampling of virus types. However, a look at the paper cited (#43) shows the strains were from multiple VHS outbreaks at several different locations over the 4 years. Revise wording to indicate the strains were from multiple outbreaks, because the broader sources of the virus strains tested makes the work much more significant. Also, I would say that they were all from cultured flounder (I believe none were from wild fish?).

2. Results section 1, lines 8-10. Here the description of defining the virulence phenotypes is not quite clear. It is important that readers understand the virulence trial was conducted 4 times and the average of the mortality observed in all trials was used to define your phenotypes. I suggest revising to "Based on the average mortality observed in four independent experiments, we selected 7 VHSV strains that caused 70-75% cumulative mortality (high virulence strains) and 6 VHSV strains that caused 35-60% cumulative mortality (low virulence strains) ....

Please check on the number of low virulence strains - that sentence in the text now says 5 low virulence strains, but in Table 1 I think there are 6. In Table 1 I would suggest adding another column on the right to indicate the virulence phenotype (high, moderate, low) - this is to help readers easily follow which strains you have defined as being in each group.

3. Results section 5 on Stability of PP55L in VHSV during viral replication. Here I think the notation should be changed because the serial passage study involves naturally occurring high and low virulence strains that differ at P55, not mutated strains. This is really testing variants PP55 and PL55, and we do not know which was ancestral. The notation PP55L indicates a mutation from P to L, which is appropriate for the in vitro mutagenesis of the infectious clone, where the original clone had a P and it was mutated to L. But the natural strains should be presented as variants without assuming you know the direction of mutation that created the polymorphism (either could have been ancestral). To make this clear in the discussion I would suggest revising Discussion line 7 to "...this single amino acid polymorphism was stably maintained for 30 passages of naturally occurring variant strains."

4. Results section 8 on Ectopic expression of PP55L generated some really fascinating results shown in Figure 5, but it seems the written text in the results section does not describe the results in the Figure fully.

a. Please explain in the methods what anti-V5 antibody is (monoclonal or polyclonal?), and say where you got it from.

b. In Fig. 5B it looks like expression of PP55L increased expression of + RNA for both the low and high virulence strains, relative to the expression levels with the pcDNA6 empty vector, but the results section mentions only an increase for the low virulence strain. Was there also an increase in expression of for the high virulence strain? This might seem logical since it would be an increased dose effect of having more PP55L (from both the high virulence strain and the ectopic expression). It seems this should have been tested statistically, and if it is significant there should be a bracket to indicate that also.

c. Figure 5C appears to show a significant decrease of (-) sense RNA expression for the high virulence strain when the PP55 is expressed ectopically, but this is not mentioned in the text of the results. That text says there was no effect, which seems to contradict Figure 5C.

d. Also in Figure 5C, as in Figure 5B, it appears that ectopic expression of PP55L results in an increase of (-) RNA expression for the high virulence strain. If so this is very interesting and should be mentioned.

5. Results section 9 on IFN genes:

a. In line 2 for the general phenomenon of viruses modulating immune responses the reviews cited are both generally for mammalian viruses. It would be useful to also cite the Purcell et al. review on Immunity to Fish Rhabdoviruses, 2012, since that is specific to the system being presented.

b. In the next sentence, also in line 2, I would suggest revising "The P protein of rhabdoviruses ..." to "The P protein of rabies virus ...". This is because the references cited for the role of the P protein (#33-37) are all for rabies virus, but the P protein of the other model rhabdovirus, vesicular stomatitis virus, does not have the same function. Therefore, to avoid suggesting that all rhabdoviruses use the same mechanism to modulate IFN, simply revise wording to "The P protein of rabies virus...".

c. In lines 10-12 the sentence starting with "However, at 12 h and 24 h ..." does not describe Figure 6A-6C quite accurately. At 12 h the 2 strains are very similar for all 3 genes, but by 24 h there is an interesting shift to lower expression for all 3 genes in rVHSV-P infected cells, with significantly less for ISG15 and Mx (if I understand the statistical notation correctly). Revise wording to present the data at 24 h more clearly.

d. second paragraph in Results section 9, line 3 says that samples were collected from i.p. infected flounder at 7 days. There is not 7 d time point shown in Figure 6?

e. second paragraph, lines 6-8, sentence starting "However, from 1 day after viral infection ...". This sentence is confusing and does not seem to describe the results in Fig. 6D and 6E well. It refers to expression of IFN1 and Mx, but there is no data for IFN1 in those figure panels. Also it says from day 1 the expression levels in rVHSV-P infected fish are "similar to those found in rVHSV-wild-infected fish", but the figure looks like it is still significantly less in 3 or 4 out of 6 panels.

6. Figure and legend suggestions:

Figure 1 - change Y axis to a maximum of 100% (there is no 120% mortality). Also indicate in the legend which experiment this is (I think it's experiment 2?), so readers can link the graph with table 1.

Figure 3 - either make the Y axes the same (both 100% maximum?), or note in the legend that the 2 panels have different Y axes.

Figure 4 - to help readers follow, note which virus strain is low virulence and high virulence. Note that the negative-sense RNA is in panels B and D, and positive-sense in A and C. then please clarify what is being compared to produce the statistical annotation - comparison of the 2 virus strains within each time point?

Figure 6 please clarify what the significance levels mean in the figure. i.e. what is compared in each case? Fold-change over mock? Or Differences between the 2 virus strains within each time point?

7. Discussion details:

a. paragraph 2, line 7 says rVHSV-P blocked induction of IFN1, ISG15, and Mx "more effectively" than rVHSV-wild. Do you have evidence to show that rVHSV-wild also blocks induction? If not, better wording might be that rVHSV-P blocks induction "relative to" rVHSV-wild.

b. paragraph 3, line 18. I think perhaps you mean "no inhibitors" specific to flounderp38MAPK and PKA?

c. paragraph 4, line 6. Since you are referring here to naturally occurring variation among strains, again I would suggest changing the notation from Pp55L to just PL55, for the reason described above in point 3.

8. Methods section for Pathogenicity Experiments in Flounder. This seems to describe one experiment, but the results section presents 4 experiments. The fish weight is important for each experiment, and it could not have all been exactly 31.85g. How were the 4 experiments done? Did they all use the same lot of fish, or were they different lots of fish? Different sizes? same challenge does in all? Also please add that the challenge period was 14 days.

PLOS authors have the option to publish the peer review history of their article (what does this mean?). If published, this will include your full peer review and any attached files.

Reviewer #1: No

Reviewer #2: No
---

## [Editor Report · Decision Letter 1]

3 Dec 2020

Dear Mr Park,

We are pleased to inform you that your manuscript 'Naturally occurring substitution in one amino acid in VHSV phosphoprotein enhances viral virulence in flounder' has been provisionally accepted for publication in PLOS Pathogens.

We appreciate the authors' good faith effort to answer all the reviewers' comments as far as the reagents will allow them to. The harmonization of the data interpretation to be more in line with what is actually presented. The explication and the addition of more rigorous statistical analyses improves upon the paper. 

Best regards,

Benhur Lee

Section Editor

PLOS Pathogens

Benhur Lee

Section Editor

PLOS Pathogens

Kasturi Haldar

Editor-in-Chief

PLOS Pathogens

orcid.org/0000-0001-5065-158X

Michael Malim

Editor-in-Chief

PLOS Pathogens

orcid.org/0000-0002-7699-2064
---

## [Editor Report · Acceptance letter]

12 Jan 2021

Dear Mr Park,

We are delighted to inform you that your manuscript, "Naturally occurring substitution in one amino acid in VHSV phosphoprotein enhances viral virulence in flounder," has been formally accepted for publication in PLOS Pathogens.

Best regards,

Kasturi Haldar

Editor-in-Chief

PLOS Pathogens

orcid.org/0000-0001-5065-158X

Michael Malim

Editor-in-Chief

PLOS Pathogens

orcid.org/0000-0002-7699-2064